# Cytoplasmic sharing through apical membrane remodeling

**Nora G Peterson[1], Benjamin M Stormo[1], Kevin P Schoenfelder[2], Juliet S King[3], Rayson RS Lee[4], Donald T Fox[1,2,3]***

[1]Department of Cell Biology, Duke University Medical Center, Durham, United States; [2]University Program in Genetics and Genomics, Duke University, Durham, United States; [3]Department of Pharmacology & Cancer Biology, Duke University Medical Center, Durham, United States; [4]Duke-NUS Medical School, Singapore, Singapore

**Abstract** Multiple nuclei sharing a common cytoplasm are found in diverse tissues, organisms, and diseases. Yet, multinucleation remains a poorly understood biological property. Cytoplasm sharing invariably involves plasma membrane breaches. In contrast, we discovered cytoplasm sharing without membrane breaching in highly resorptive *Drosophila* rectal papillae. During a six-hour developmental window, 100 individual papillar cells assemble a multinucleate cytoplasm, allowing passage of proteins of at least 62 kDa throughout papillar tissue. Papillar cytoplasm sharing does not employ canonical mechanisms such as incomplete cytokinesis or muscle fusion pore regulators. Instead, sharing requires gap junction proteins (normally associated with transport of molecules < 1 kDa), which are positioned by membrane remodeling GTPases. Our work reveals a new role for apical membrane remodeling in converting a multicellular epithelium into a giant multinucleate cytoplasm.

*For correspondence:
don.fox@duke.edu

**Competing interests:** The authors declare that no competing interests exist.

## Introduction

Throughout the tree of life, there are upper limits to the size of individual cells. This size limitation is imposed by genome content, which impacts biosynthetic capacity and cell growth (*Conlon and Raff, 1999*; *Mueller, 2015*). In diverse tissues and organisms, the existence of 'giant cells' is driven by polyploidy, the presence of greater than a diploid genome content (*Van de Peer et al., 2017*; *Schoenfelder and Fox, 2015*). Purposes of polyploidy across evolution remain largely unknown. However, one potential advantage of a tissue containing few, large polyploid cells vs. numerous, small diploid cells is the ability of cytoplasmic components to move over much larger distances.

A common form of polyploidy is multinucleation. Sharing of cytoplasm in a multinucleate tissue or organism is an important and recurring adaptation across evolution. Multinucleate cells can be large, metabolically-active cells with unique shapes and functions ranging from specialized force distribution to tissue barrier preservation. During organismal development, examples of multinucleation include animal skeletal muscle, mammalian osteoclasts, and mammalian syncytial placental trophoblasts (*Deng et al., 2017*; *Gerbaud and Pidoux, 2015*; *Pereira et al., 2018*). Multinucleation also arises in response to tissue stress, such as following injury to the *Drosophila* abdominal epithelium or the human corneal epithelium (*Losick et al., 2013*; *Ikebe et al., 1986*). A commonality of these numerous examples of multinucleation is the ability to exchange, over long distances, cytoplasmic components such as RNA, proteins, and even organelles (*Rustom et al., 2004*; *McLean and Cooley, 2013*).

The cellular mechanisms underlying multinucleation are diverse. During cell division, multinucleation can occur through incomplete cytokinesis, followed by formation of a stable cytoplasmic bridge between nuclei. This process occurs in diverse examples of germ cell development

**eLife digest** Most cells are self-contained – they have a cell membrane that delimits and therefore defines the cell, separating it from other cells and from its environment. But sometimes several cells interconnect and form collectives so they can pool their internal resources. Some of the best-known examples of this happen in animal muscle cells and in the placenta of mammals. These cell collectives share their cytoplasm – the fluid within the cell membrane that contains the cell organelles – in one of two ways. Cells can either remain linked instead of breaking away when they divide, or they can fuse their membranes with those of their neighbors. Working out how cells link to their neighbors is difficult when so few examples of cytoplasm sharing are available for study. One way to tackle this is to try and find undiscovered cell collectives in an animal that is already heavily studied in the lab, such as the fruit fly *Drosophila melanogaster*.

Peterson et al. used a genetic system that randomly labels each cell of the developing fly with one of three fluorescent proteins. These proteins are big and should not move between cells unless they are sharing their cytoplasm. This means that any cell containing two or more different colors of fluorescent protein must be connected to at least one of its neighbors. The experiment revealed that the cells of the fruit fly rectum share their cytoplasm in a way never seen before. This sharing occurs at a consistent point in the development of the fruit fly and uses a different set of genes to those used by interconnecting cells in mammal muscles and placenta. These genes produce proteins that reshape the membranes of the cells and fit them with gap junctions – tiny pores that cross from one membrane to the next, allowing the passage of very small molecules. In this case, the gap junctions allowed the cells to share molecules much larger than seen before. The result is a giant cell membrane containing the cytoplasm and organelles of more than a hundred individual cells.

These findings expand scientists' understanding of how cells in a tissue can share cytoplasm and resources. They also introduce a new tissue in the fruit fly that can be used in future studies of cytoplasm sharing. Relatives of fruit flies, including fruit pests and mosquitos, have similar cell structure to the fruit fly, which means that further investigations using this system could result in advances in agriculture or human health.

(*Greenbaum et al., 2011*) and also in some somatic cells such as the ring canal of the *Drosophila* ovary (*McLean and Cooley, 2013*) and the plasmodesmata of plants (*Lůcas and Wolf, 1993*). A second major mechanism of multinucleation involves plasma membrane breaches. Such breaches can involve distinct actin-based protrusive structures. Podosome-like structures facilitate multinucleation in *Drosophila* skeletal muscle and mammalian macrophages (*Faust et al., 2019*; *Sens et al., 2010*). While the mechanisms are diverse, one common feature of the above-discussed examples of multinucleation and cytoplasm sharing identified to date are clearly visible plasma membrane disruptions.

Here, we report a visual animal-wide screen, using multi-color lineage labeling approaches in the tractable animal model *Drosophila melanogaster,* for multinucleate tissues that share cytoplasm. We discover cytoplasm sharing in the rectal papilla, a common insect resorptive intestinal epithelium that is critical for maintaining ionic homeostasis (*Wigglesworth, 1932*; *Cohen et al., 2020*). Likely due to its extreme proximal location in the gut of many insect species, this epithelium is linked to the infiltration of diverse pathogens, such as those involved in African sleeping sickness and also viruses being pursued as insect control measures (*Gu et al., 2010*; *Filosa et al., 2019*). Here, we reveal that cytoplasm sharing onset in *Drosophila* papillae occurs during a short developmental window, indicating robust molecular regulation. We find that papillar cytoplasm sharing requires neither incomplete cytokinesis nor canonical actin-based membrane breach regulators. Using transmission electron microscopy, we further identify that this developmentally programmed process involves extensive remodeling of apical junctions and lateral membranes, but not clearly identifiable plasma membrane breaches. Using genetic screening, we implicate specific regulators of membrane remodeling, notably the GTPase Dynamin/Shibire, in the mechanism of papillar cytoplasmic sharing. From analysis of *shibire* mutants, we uncover a requirement for gap junction establishment and specific gap junction proteins in papillar cytoplasm sharing. Mutant animals defective in papillar cytoplasm sharing are intolerant of a high-salt diet, indicating a physiological role of long-range cytoplasm movement in this tissue. Unlike all known examples of multinucleation, our results show that

cytoplasm sharing in rectal papillae requires developmentally programmed apical membrane remodeling, which creates a giant resorptive epithelial network of 100 nuclei. This tissue represents a new system to investigate the diversity of multicellular tissue organization and mechanisms and functions of cytoplasm sharing.

## Results

### *Drosophila* hindgut papillae undergo developmentally programmed cytoplasmic sharing

To identify new examples of adult tissues in *Drosophila* that share cytoplasm, we ubiquitously expressed *Cre* and *UAS-dBrainbow* (*Hampel et al., 2011*; *Figure 1A*), a Cre-Lox-based system that randomly labels cells with only one of three fluorescent proteins. We used animals heterozygous for *UAS-dBrainbow* to ensure single-labeling of cells. We ubiquitously expressed *Cre,* which does not require heat-shock induction, from early embryonic stages (before cells endocycle to become polyploid). Cre-mediated excision occurs independently of Gal4 expression and Gal80$^{ts}$ repression of dBrainbow. Therefore, we can ensure that multi-labeled cells only arise by cytoplasm sharing between cells not related by cell division or incomplete cytokinesis (*Figure 1B*). We examined a wide range of tissues (*Figure 1—figure supplement 1A*). From our screen, we discovered that the rectal papilla is a new example of a tissue with cytoplasm sharing. Adult *Drosophila* contain four papillae, each with 100 nuclei of genome content between 8 and 16C (*Fox et al., 2010*), that reside in the posterior hindgut (*Figure 1C*). Each papilla is a polarized epithelial cone with the apical region facing the gut lumen and the basal region surrounding a central canal that connects to the fly's hemolymph (*Figure 1D*). The papillar structure supports its function to reabsorb water, ions, and small molecules from the gut lumen and recycle them back to the hemolymph (*Cohen et al., 2020*). Knowing that adult papillar cells share cytoplasm, we next used our dBrainbow system to identify when papillar cells begin to share relative to other developmental events that we previously identified (*Figure 1E*). Using both fixed and live imaging of whole organs, we found that at 62 hours post-puparium formation (HPPF), each papillar cell contains only one dBrainbow label (*Figure 1F*). By contrast, at 69HPPF, multi-labeled cells are apparent (*Figure 1F',H–H'*). We quantitatively measured papillar sharing across the tissue (*Figure 1—figure supplement 1B*, Materials and methods) and found that cytoplasm sharing initiates over a narrow 6 hr period (68-74HPPF, *Figure 1G*). Our results suggested that at least RNA and possibly protein passes between papillar cells to facilitate cytoplasm sharing. To directly test if protein is shared, we photo-activated GFP (GFP$^{PA}$) in single adult papillar cells and observed in real time whether GFP$^{PA}$ spreads to adjacent cells. We find the principal papillar cells, but not the secondary cells at the papillar base (*Garayoa et al., 1999*; *Figure 1—figure supplement 1C*), share protein across an area of at least several nuclei (*Figure 1I–I'*). We next tested whether a larger protein can be shared between papillar cells. We used rectal papillae RNA-sequencing data (*Leader et al., 2018*) to identify proteins that are endogenously expressed, cytoplasmic, and relatively large. We therefore generated flies expressing a UAS-inducible, photoactivatable GFP fused to *Glyceraldehyde 3 phosphate dehydrogenase 2* (*UAS-Gapdh2-GFP$^{PA}$*). This construct should produce a tagged protein of 62.3 kDa. We found that Gapdh2-GFP$^{PA}$ protein is shared between cells, as it never stops at a papillar cell–cell boundary, though it may move at a slower rate than GFP$^{PA}$ (*Figure 1—figure supplement 1D*). Therefore, proteins as large as ~62 kDa (the size of GFP-tagged Gapdh2) can move across an area covered by multiple papillar nuclei. Additionally, the movement of our Gapdh2 transgenic protein indicates that papillar cells likely share endogenously expressed proteins. These results indicate that papillae undergo a developmentally programmed conversion from 100 individual cells to a single giant multinuclear cytoplasm that shares the products of ~1200 genomes.

We next examined whether cytoplasm sharing requires either programmed endocycles or mitoses. We have previously shown that larval papillar cells first undergo endocycles, which increase cellular ploidy. Then, during metamorphosis, pupal papillar cells disassemble polytene chromosomes and undergo polyploid mitotic cycles, which increase cell number (*Fox et al., 2010*; *Stormo and Fox, 2016*; *Stormo and Fox, 2019*). Both endocycles and mitoses occur well prior to the start of papillar cytoplasm sharing (*Figure 1E*). Papillar endocycles require the Anaphase-Promoting Complex/Cyclosome regulator *fizzy-related* (*fzr*) while the papillar mitoses require Notch signaling

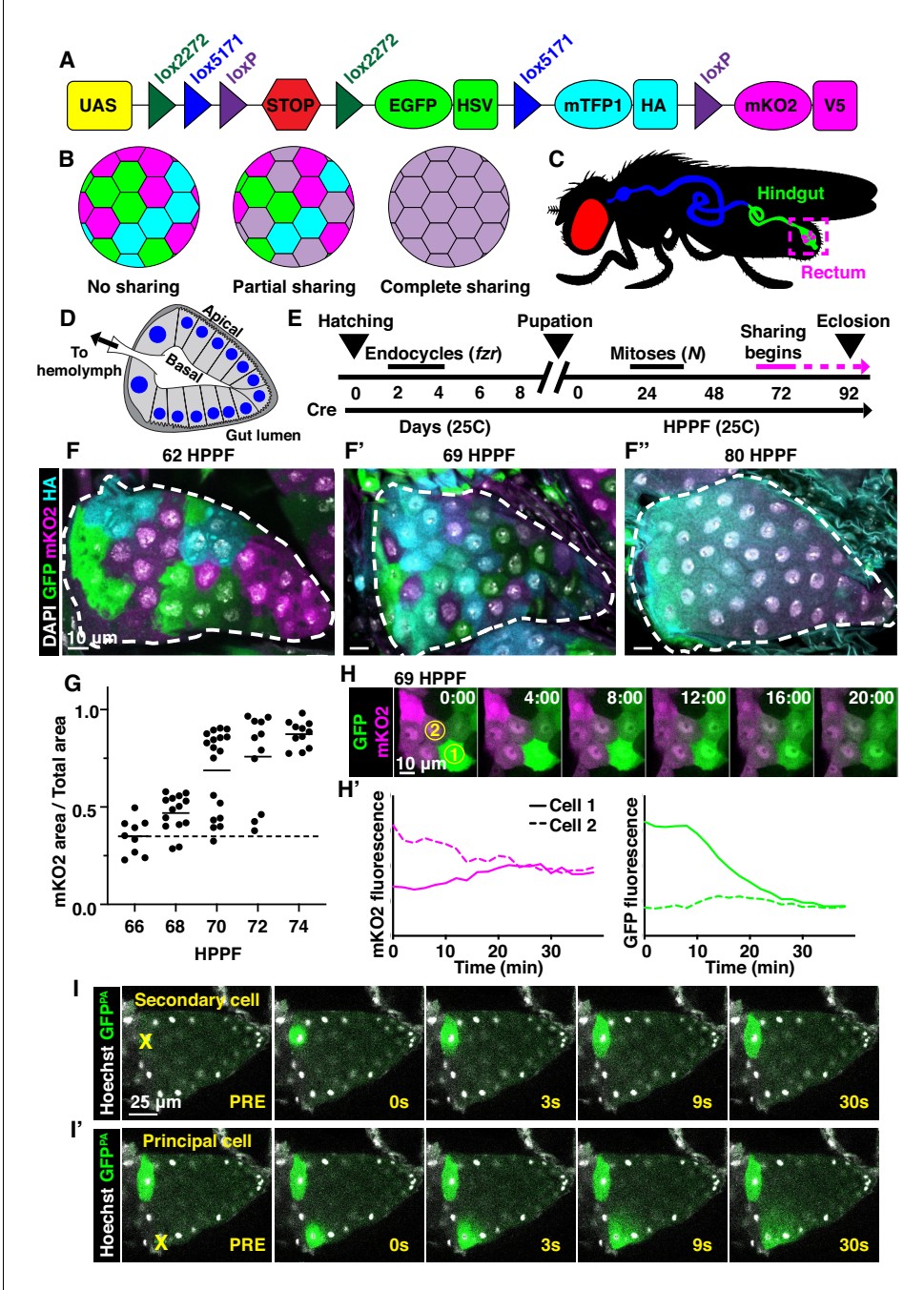

**Figure 1.** Developmentally programmed cytoplasmic sharing in *Drosophila* papillae. (A) The dBrainbow construct (*Hampel et al., 2011*). Cre recombinase randomly excises one pair of lox sites, and approximately 1/3 of cells express either EGFP, mKO2, or mTFP1. (B) Model of dBrainbow expression with no, partial, or complete cytoplasmic sharing. (C) *Drosophila* digestive tract with rectum containing four papillae labeled in magenta box. (D) Cartoon of a cross-section through an adult rectal papilla. The papilla consists of an epithelial cone with the apical region facing the gut lumen and the interior basal region facing a central canal leading to the fly hemolymph. The principal papillar cells have microvilli-like projections on the apical edge. One layer of larger, secondary cells forms the base of the papilla. The papilla is covered in a cuticle layer (dark gray). Nuclei are marked in blue. (E) Approximate timeline of ubiquitous Cre induction and cytoplasm sharing onset (68–74 HPPF) within papillar development (*Fox et al., 2010*). Cytoplasmic sharing is temporally separate from papillar mitoses. (F–F'') Representative *dBrainbow* papillae at 62 (F), 69 (F'), or 80 (F'') hours post-puparium formation (HPPF). (G) Cytoplasmic sharing quantification during pupal development. Lines = mean at each time, which differs

*Figure 1 continued on next page*

*Figure 1 continued*

significantly between 66 and 74 HPPF (p<0.0001). Each point = 1 animal (N = 9–18, rep = 2). (**H**) Live *dBrainbow*-labeled papillar cells during cytoplasmic sharing (69 HPPF). (**H'**) Fluorescence of neighboring cells in (**H**). (**I–I'**) Representative adult papilla expressing photo-activatable GFP (GFP^PA). Single cells were photo-activated (yellow X) in secondary cells (**I**) and principal cells (**I'**). Time = seconds after activation.

The online version of this article includes the following figure supplement(s) for figure 1:

**Figure supplement 1.** The hindgut rectal papillae share cytoplasm independent of mitosis.

---

(*Schoenfelder et al., 2014*). Knockdown of *fzr* significantly disrupts cytoplasm sharing (*Figure 1—figure supplement 1E,F,H*). We hypothesize that endocycles are required for differentiation of the papillae, which later enables these cells to trigger cytoplasm sharing. In contrast, blocking Notch signaling, which initiates papillar mitotic divisions (*Fox et al., 2010*), does not prevent sharing (*Figure 1—figure supplement 1E,G,H*). Thus, papillar cytoplasm sharing requires developmentally programmed endocycles but not mitotic cycles.

## Cytoplasmic sharing requires membrane remodeling proteins

As our *dBrainbow* approach only identifies cytoplasm sharing events that do not involve incomplete division/cytokinesis, we examined whether sharing results from fusion pore formation, as in skeletal muscle. A well-studied model of such cell–cell fusion in *Drosophila* is myoblast fusion, which requires an actin-based podosome (*Richardson et al., 2007*; *Sens et al., 2010*). We conducted a candidate *dBrainbow*-based RNAi screen (77 genes, *Figure 2A*, *Table 1*) of myoblast fusion regulators and other plasma membrane components. Remarkably, 0/15 myoblast fusion genes from our initial screen regulate papillar cytoplasm sharing (*Figure 2A*, *Figure 2—figure supplement 1A*, *Table 1*). Furthermore, dominant-negative forms of Rho family GTPases have no impact on *dBrainbow* labeling (*Figure 2—figure supplement 1B*), providing additional evidence against actin-based cytoplasm sharing. Instead, we found 8/77 genes, including subunits of the vacuolar H+ ATPase (*Vha16-1*), ESCRT-III complex (*Vps2*), and exocyst (*Exo84*) (*Figure 2A*) are required for papillar cytoplasm sharing. Through additional screening, the only myoblast fusion regulator required for papillar cytoplasm sharing is *singles bar* (*sing*), a presumed vesicle trafficking gene (*Estrada et al., 2007*; *Figure 2—figure supplement 1A*). Given the enrichment of our candidate screen hits in membrane trafficking and not myoblast fusion, we further explored the role of membrane trafficking in cytoplasm sharing.

We conducted two secondary *dBrainbow* screens to find specific membrane trafficking pathway components that regulate papillar sharing. First, a focused candidate membrane trafficking screen revealed additional components (12/36 genes screened, *Figure 2B*, *Table 2*) including three more vacuolar H+ ATPase subunits, five more exocyst components, and the Dynamin GTPase *shibire* (*shi*) (*Figure 2B,D,E,H*). Second, we screened constitutively-active and dominant-negative versions of all 31 *Drosophila* Rabs. Sharing requires only a small number of Rabs, specifically the ER/Golgi-associated *Rab1*, the early endosome-associated *Rab5,* and the recycling endosome-associated *Rab11* (*Figure 2C,D,F–H*). Given our identification of the membrane vesicle recycling circuit involving *shi*, *Rab5*, and *Rab11*, we focused on these genes. Two unique RNAi lines for each gene show consistent sharing defects, and most of these knockdowns completely recapitulate the pre-sharing state (*Figure 2H*). Despite exhibiting strong cytoplasm sharing defects, *shi*, *Rab5*, and *Rab11 RNAi* papillae appear morphologically normal, with only minor cell number decreases (*Figure 2—figure supplement 1C*). These results suggest that membrane recycling GTPases regulate a specific developmental event associated with cytoplasm sharing, and not papillar morphogenesis. In agreement with these GTPases acting during development, rather than as part of an ongoing transport process, GTPase knockdown after sharing onset does not block cytoplasm sharing (*Figure 2—figure supplement 1D–F*). Together, our screens reveal that membrane trafficking, particularly Dynamin-mediated endocytosis and early/recycling endosome trafficking, regulates papillar cytoplasmic sharing.

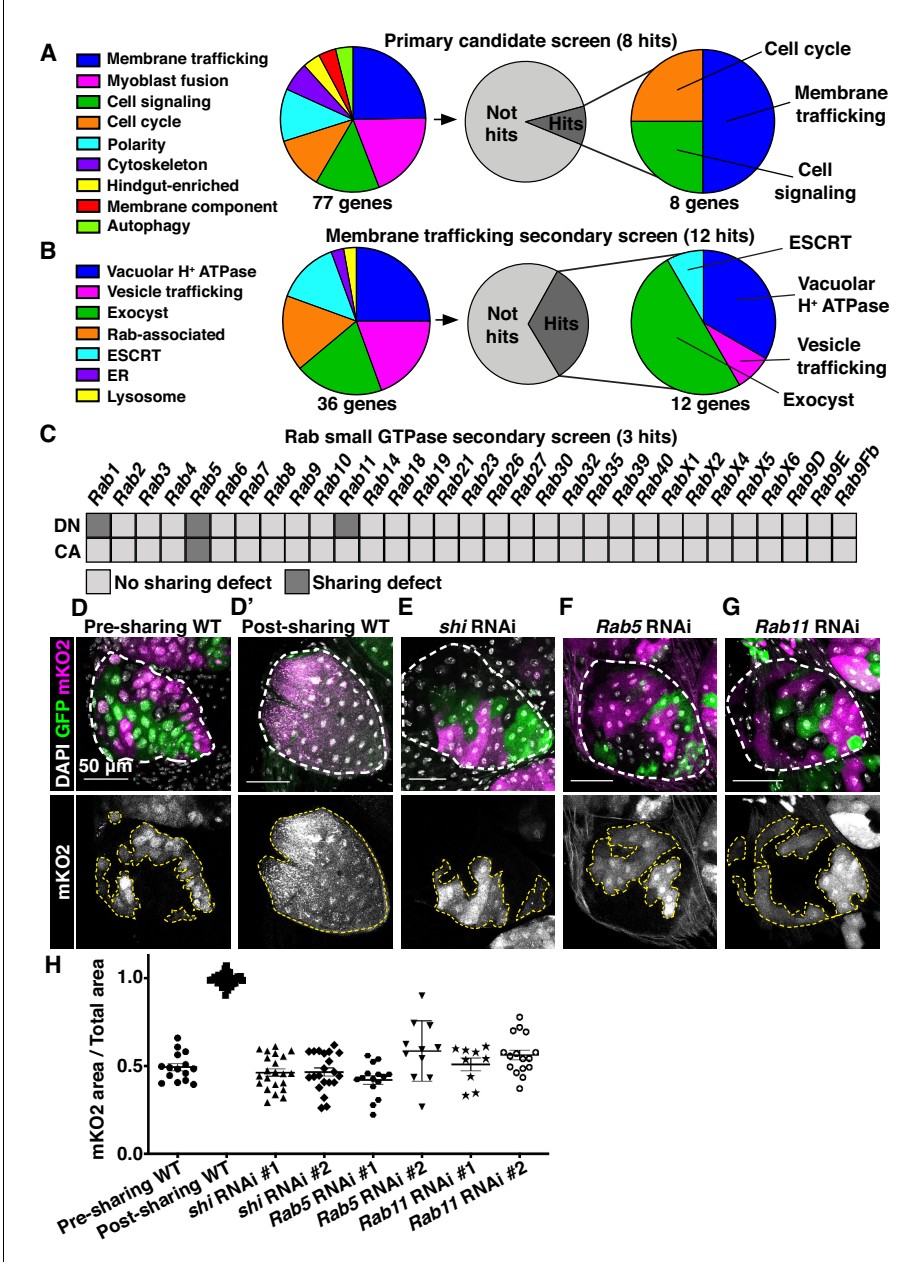

**Figure 2.** Cytoplasmic sharing requires membrane remodeling proteins. (**A**) Primary *dBrainbow* candidate screen. RNAi and dominant-negative versions of 77 genes representing the indicated roles were screened for sharing defects, and eight genes were identified. (**B**) Secondary membrane trafficking screen. 36 genes were screened with 12 sharing genes identified. (**C**) Secondary screen of dominant-negative and constitutively-active Rab GTPases. (**D–G**) Representative *dBrainbow* in (**D–D'**) wild type (WT) (**D**) pre-sharing (48HPPF) and (**D'**) post-sharing (young adults), (**E**) adult *shi RNAi*, (**F**) adult *Rab5 RNAi*, (**G**) adult *Rab11 RNAi*. (**H**) Quantification of (**D–G**), including two RNAi lines for *shi*, *Rab5*, and *Rab11*. Pre-sharing and knock downs differ significantly from post-sharing WT (p<0.0001, N = 9–32, rep = 2–3).

The online version of this article includes the following figure supplement(s) for figure 2:

**Figure supplement 1.** Membrane trafficking genes expressed during a developmental window regulate cytoplasm sharing.

**Table 1.** Cytoplasm sharing primary candidate screen gene results.

| Gene category | Gene | Annotation symbol | Gene ID | Sharing disrupted? |
|---|---|---|---|---|
| Autophagy | Atg1 | CG10967 | FBgn0260945 | No |
| Autophagy | Atg7 | CG5489 | FBgn0034366 | No |
| Autophagy | Atg8a | CG32672 | FBgn0052672 | No |
| Cell cycle/Chromosomes | blue | NA | FBgn0283709 | No |
| Cell cycle/Chromosomes | CapD2 | CG1911 | FBgn0039680 | No |
| Cell cycle/Chromosomes | Cdc2 | CG5363 | FBgn0004106 | Yes |
| Cell cycle/Chromosomes | Clamp | CG1832 | FBgn0032979 | No |
| Cell cycle/Chromosomes | endos | CG6513 | FBgn0061515 | No |
| Cell cycle/Chromosomes | fzr | CG3000 | FBgn0262699 | Yes |
| Cell cycle/Chromosomes | Mi-2 | CG8103 | FBgn0262519 | No |
| Cell cycle/Chromosomes | Rbp9 | CG3151 | FBgn0010263 | No |
| Cell cycle/Chromosomes | SA-2 | CG13916 | FBgn0043865 | No |
| Cell signaling | Chico | CG5686 | FBgn0024248 | No |
| Cell signaling | Egfr | CG10079 | FBgn0003731 | Yes |
| Cell signaling | grk | CG17610 | FBgn0001137 | No |
| Cell signaling | N | CG3936 | FBgn0004647 | No |
| Cell signaling | Ptp61F | CG9181 | FBgn0267487 | No |
| Cell signaling | rho | CG1004 | FBgn0004635 | Yes |
| Cell signaling | ru | CG1214 | FBgn0003295 | No |
| Cell signaling | spi | CG10334 | FBgn0005672 | No |
| Cell signaling | stet | CG33166 | FBgn0020248 | No |
| Cell signaling | wts | CG12072 | FBgn0011739 | No |
| Cell signaling | βggt-II | CG18627 | FBgn0028970 | No |
| Cytoskeleton | ALiX | CG12876 | FBgn0086346 | No |
| Cytoskeleton | Cdc42 | CG12530 | FBgn0010341 | No |
| Cytoskeleton | DCTN1-p150 | CG9206 | FBgn0001108 | No |
| Cytoskeleton | pav | CG1258 | FBgn0011692 | No |
| Cytoskeleton | wash | CG13176 | FBgn0033692 | No |
| Hindgut-enriched | dac | CG4952 | FBgn0005677 | No |
| Hindgut-enriched | Dr | CG1897 | FBgn0000492 | No |
| Hindgut-enriched | nrv3 | CG8663 | FBgn0032946 | No |
| Membrane component | Flo1 | CG8200 | FBgn0024754 | No |
| Membrane component | Flo2 | CG32593 | FBgn0264078 | No |
| Membrane component | Iris | CG4715 | FBgn0031305 | No |
| Myoblast fusion | Arf51F | CG8156 | FBgn0013750 | No |
| Myoblast fusion | Arp2 | CG9901 | FBgn0011742 | No |
| Myoblast fusion | Arp3 | CG7558 | FBgn0262716 | No |
| Myoblast fusion | Ced-12 | CG5336 | FBgn0032409 | No |
| Myoblast fusion | dock | CG3727 | FBgn0010583 | No |
| Myoblast fusion | hbs | CG7449 | FBgn0029082 | No |
| Myoblast fusion | Hem | CG5837 | FBgn0011771 | No |
| Myoblast fusion | mbc | CG10379 | FBgn0015513 | No |
| Myoblast fusion | Rac1 | CG2248 | FBgn0010333 | No |
| Myoblast fusion | Rho1 | CG8416 | FBgn0014020 | No |

*Table 1 continued on next page*

*Table 1 continued*

| Gene category | Gene | Annotation symbol | Gene ID | Sharing disrupted? |
|---|---|---|---|---|
| Myoblast fusion | *rols* | CG32096 | FBgn0041096 | No |
| Myoblast fusion | *rst* | CG4125 | FBgn0003285 | No |
| Myoblast fusion | *SCAR* | CG4636 | FBgn0041781 | No |
| Myoblast fusion | *siz* | CG32434 | FBgn0026179 | No |
| Myoblast fusion | *WASp* | CG1520 | FBgn0024273 | No |
| Polarity | *Abi* | CG9749 | FBgn0020510 | No |
| Polarity | *CadN* | CG7100 | FBgn0015609 | No |
| Polarity | *cindr* | CG31012 | FBgn0027598 | No |
| Polarity | *cno* | CG42312 | FBgn0259212 | No |
| Polarity | *Gli* | CG3903 | FBgn0001987 | No |
| Polarity | *l(2)gl* | CG2671 | FBgn0002121 | No |
| Polarity | *Nrg* | CG1634 | FBgn0264975 | No |
| Polarity | *sdt* | CG32717 | FBgn0261873 | No |
| Polarity | *shg* | CG3722 | FBgn0003391 | No |
| Vesicle trafficking | *Atl* | CG6668 | FBgn0039213 | No |
| Vesicle trafficking | *Bet1* | CG14084 | FBgn0260857 | No |
| Vesicle trafficking | *Chmp1* | CG4108 | FBgn0036805 | No |
| Vesicle trafficking | *CHMP2B* | CG4618 | FBgn0035589 | No |
| Vesicle trafficking | *dnd* | CG6560 | FBgn0038916 | No |
| Vesicle trafficking | *Exo84* | CG6095 | FBgn0266668 | Yes |
| Vesicle trafficking | *lerp* | CG31072 | FBgn0051072 | No |
| Vesicle trafficking | *Rab11* | CG5771 | FBgn0015790 | Yes |
| Vesicle trafficking | *Rab23* | CG2108 | FBgn0037364 | No |
| Vesicle trafficking | *Rab4* | CG4921 | FBgn0016701 | No |
| Vesicle trafficking | *Rab7* | CG5915 | FBgn0015795 | No |
| Vesicle trafficking | *Rab8* | CG8287 | FBgn0262518 | No |
| Vesicle trafficking | *RabX4* | CG31118 | FBgn0051118 | No |
| Vesicle trafficking | *Vha16-1* | CG3161 | FBgn0262736 | Yes |
| Vesicle trafficking | *Vha55* | CG17369 | FBgn0005671 | No |
| Vesicle trafficking | *VhaAC39-1* | CG2934 | FBgn0285910 | No |
| Vesicle trafficking | *VhaAC39-2* | CG4624 | FBgn0039058 | No |
| Vesicle trafficking | *Vps2* | CG14542 | FBgn0039402 | Yes |
| Vesicle trafficking | *Vps33b* | CG5127 | FBgn0039335 | No |
| Total screen results | | | | |
| Sharing disrupted | 8 | | | |
| No sharing phenotype | 69 | | | |
| Total | 77 | | | |
| Screen results by category | | | | |
| Polarity | 9 | | | |
| Vesicle trafficking | 19 | | | |
| Myoblast fusion | 15 | | | |
| Cell cycle/Chromosomes | 9 | | | |
| Cell signaling | 11 | | | |
| Autophagy | 3 | | | |

*Table 1 continued on next page*

*Table 1 continued*

| Gene category | Gene | Annotation symbol | Gene ID | Sharing disrupted? |
|---|---|---|---|---|
| Cytoskeleton | 5 | | | |
| Hindgut-enriched | 3 | | | |
| Membrane component | 3 | | | |
| Total | 77 | | | |

## Gap junction establishment, but no membrane breaches, accompany cytoplasm sharing

To better understand how membrane trafficking GTPases initiate cytoplasm sharing during development, we examined endosome and Shi localization during sharing onset. We imaged a GFP-tagged pan-endosome marker (*myc-2x-FYVE*), overexpression of which should not alter endosome shape or localization (*Gillooly et al., 2000*; *Wucherpfennig et al., 2003*), and a Venus-tagged *shi* before and after sharing. Endosomes are evenly distributed shortly before sharing, but become highly polarized at the basal membrane around the time of sharing onset (*Figure 3A–A',C*, *Figure 3—figure supplement 1A*). This basal endosome repositioning requires Shi (*Figure 3B–C*, *Figure 3—figure supplement 1A*) and the change in endosome localization is attributed to Rab5-positive early endosomes (*Figure 3—figure supplement 1B–B''*). Additionally, Shi localization changes from apical polarization to a uniform distribution during sharing onset (*Figure 3D–E*). These localization changes indicate that membrane trafficking factors which regulate cytoplasm sharing are highly dynamic during cytoplasm sharing onset.

To determine what membrane remodeling events underlie GTPase-dependent cytoplasm sharing, we turned to ultrastructural analysis. Adult ultrastructure and physiology of papillar cells has been examined previously in *Drosophila* (*Wessing and Eichelberg, 1973*) and related insects (*Gupta and Berridge, 1966*). These cells contain elaborate membrane networks that facilitate selective ion resorption from the gut lumen, facing the apical side of papillar cells, to the hemolymph, facing the basal side. Still, little is known about developmental processes or mechanisms governing the unique papillar cell architecture. We looked for changes in cell–cell junctions and lateral membranes that coincide with cytoplasm sharing, especially to determine if there is a physical membrane breach between cells. We identified several dramatic changes in membrane architecture. First, apical microvilli-like structures form during sharing onset (*Figure 3F–F''*). Just basal to the microvilli, apical cell–cell junctions are straight in early pupal development and compress into a more curving, tortuous morphology around the time of cytoplasm sharing onset (*Figure 3—figure supplement 1C–C''*). One of the most striking changes, coincident with Shi re-localization, is formation of pan-cellular endomembrane stacks surrounding mitochondria. These stacks are likely sites for active ion transport, such as that mediated by the P-type $Na^+/K^+$-ATPase, coupled to mitochondria for ATP (*Figure 3G–G''*; *Berridge and Gupta, 1967*; *Patrick et al., 2006*). Thus, massive apical and intracellular plasma membrane reorganization coincides with both cytoplasm sharing and Shi/endosome re-localization. We next assessed whether the extensive membrane remodeling requires Shi, Rab5, and Rab11. In *shi* and *Rab5 RNAi* animals, microvilli protrude downward, instead of upward (*Figure 3H–J*). Additionally, apical junctions do not compress as in controls (*Figure 3—figure supplement 1D–F*). Notably, membrane stacks are greatly reduced (*Figure 3K–M*). *shi RNAi* animals exhibit numerous trapped vesicles, consistent with a known role for Dynamin in membrane vesicle severing (*Damke et al., 1994*; *Hinshaw and Schmid, 1995*; *Figure 3L*, inset). Together, we find that Shi and endosomes extensively remodel membranes during papillar cytoplasm sharing.

## Gap junction proteins are required for cytoplasmic sharing

Our extensive ultrastructural analysis did not reveal any clear breaches in the plasma membrane, despite numerous membrane alterations. Adult papillae exhibit large extracellular spaces between nuclei that eliminate the possibility of cytoplasm sharing throughout much of the lateral membrane (*Figure 3—figure supplement 2A*; *Wessing and Eichelberg, 1973*; *Gupta and Berridge, 1966*). Instead, through our GTPase knockdown studies, we identified a striking alteration in the apical cell–cell interface that strongly correlates with cytoplasm sharing. Specifically, *shi* animals frequently lack

**Table 2.** Membrane trafficking primary and secondary candidate screen gene results.

| Gene category | Gene subcategory | Gene | Annotation symbol | Gene ID | Sharing disrupted? | Screen |
|---|---|---|---|---|---|---|
| Membrane trafficking | ER | Atl | CG6668 | FBgn0039213 | No | Primary |
| Membrane trafficking | ESCRT | Chmp1 | CG4108 | FBgn0036805 | No | Primary |
| Membrane trafficking | ESCRT | CHMP2B | CG4618 | FBgn0035589 | No | Primary |
| Membrane trafficking | ESCRT | lsn | CG6637 | FBgn0260940 | No | Secondary |
| Membrane trafficking | ESCRT | Vps2 | CG14542 | FBgn0039402 | Yes | Primary |
| Membrane trafficking | ESCRT | Vps4 | CG6842 | FBgn0283469 | No | Secondary |
| Membrane trafficking | Exocyst | Exo70 | CG7127 | FBgn0266667 | No | Secondary |
| Membrane trafficking | Exocyst | Exo84 | CG6095 | FBgn0266668 | Yes | Primary |
| Membrane trafficking | Exocyst | Sec10 | CG6159 | FBgn0266673 | Yes | Secondary |
| Membrane trafficking | Exocyst | Sec15 | CG7034 | FBgn0266674 | Yes | Secondary |
| Membrane trafficking | Exocyst | Sec5 | CG8843 | FBgn0266670 | Yes | Secondary |
| Membrane trafficking | Exocyst | Sec6 | CG5341 | FBgn0266671 | Yes | Secondary |
| Membrane trafficking | Exocyst | Sec8 | CG2095 | FBgn0266672 | Yes | Secondary |
| Membrane trafficking | Lysosome | lerp | CG31072 | FBgn0051072 | No | Primary |
| Membrane trafficking | Rab-associated | CG41099 | CG41099 | FBgn0039955 | No | Secondary |
| Membrane trafficking | Rab-associated | mtm | CG9115 | FBgn0025742 | No | Secondary |
| Membrane trafficking | Rab-associated | nuf | CG33991 | FBgn0013718 | No | Secondary |
| Membrane trafficking | Rab-associated | Rala | CG2849 | FBgn0015286 | No | Secondary |
| Membrane trafficking | Rab-associated | Rep | CG8432 | FBgn0026378 | No | Secondary |
| Membrane trafficking | Rab-associated | Rip11 | CG6606 | FBgn0027335 | No | Secondary |
| Membrane trafficking | Vacuolar H+ ATPase | Vha16-1 | CG3161 | FBgn0262736 | Yes | Primary |
| Membrane trafficking | Vacuolar H+ ATPase | Vha16-2 | CG32089 | FBgn0028668 | No | Secondary |
| Membrane trafficking | Vacuolar H+ ATPase | Vha16-3 | CG32090 | FBgn0028667 | No | Secondary |
| Membrane trafficking | Vacuolar H+ ATPase | Vha16-5 | CG6737 | FBgn0032294 | Yes | Secondary |
| Membrane trafficking | Vacuolar H+ ATPase | Vha55 | CG17369 | FBgn0005671 | No | Primary |
| Membrane trafficking | Vacuolar H+ ATPase | VhaAC39-1 | CG2934 | FBgn0285910 | No | Primary |
| Membrane trafficking | Vacuolar H+ ATPase | VhaAC39-2 | CG4624 | FBgn0039058 | No | Primary |
| Membrane trafficking | Vacuolar H+ ATPase | VhaPPA1-1 | CG7007 | FBgn0028662 | Yes | Secondary |
| Membrane trafficking | Vacuolar H+ ATPase | VhaPPA1-2 | CG7026 | FBgn0262514 | Yes | Secondary |
| Membrane trafficking | Vesicle trafficking | Bet1 | CG14084 | FBgn0260857 | No | Primary |
| Membrane trafficking | Vesicle trafficking | Chc | CG9012 | FBgn0000319 | No | Secondary |
| Membrane trafficking | Vesicle trafficking | dnd | CG6560 | FBgn0038916 | No | Primary |
| Membrane trafficking | Vesicle trafficking | shi | CG18102 | FBgn0003392 | Yes | Secondary |
| Membrane trafficking | Vesicle trafficking | Vps29 | CG4764 | FBgn0031310 | No | Secondary |
| Membrane trafficking | Vesicle trafficking | Vps33b | CG5127 | FBgn0039335 | No | Primary |
| Membrane trafficking | Vesicle trafficking | Vps35 | CG5625 | FBgn0034708 | No | Secondary |
| Total screen results | | | | | | |
| Sharing disrupted | 12 | | | | | |
| No sharing phenotype | 24 | | | | | |
| Total | 36 | | | | | |
| Screen results by category | Total | Hits | | | | |
| ER | 1 | 0 | | | | |
| ESCRT | 5 | 1 | | | | |
| Exocyst | 7 | 6 | | | | |

*Table 2 continued on next page*

*Table 2 continued*

| Gene category | Gene subcategory | Gene | Annotation symbol | Gene ID | Sharing disrupted? | Screen |
|---|---|---|---|---|---|---|
| Lysosome | 1 | 0 | | | | |
| Rab-associated | 6 | 0 | | | | |
| Vacuolar H+ ATPase | 9 | 4 | | | | |
| Vesicle trafficking | 7 | 1 | | | | |
| Total | 36 | | | | | |

apical gap junctions (*Figure 3N–O*) (p<0.0001) (*Figure 3P, Figure 3—figure supplement 1H–H''*). Upon closer examination of control animal development, we find that apical gap junction-like structures arise at cytoplasm sharing onset. There is almost no gap junction-like structure before cytoplasm sharing (*Figure 4A–B, Figure 2—figure supplement 1A–A''*). Given our electron micrograph results, we determined which innexins, the protein family associated with gap junctions in invertebrates (*Bauer et al., 2005; Phelan et al., 1998*), are expressed in rectal papillae. From RNA-seq data (*Methods*), we determined that *ogre* (*Inx1*), *Inx2*, and *Inx3* are most highly expressed (*Figure 4C*). This combination of innexins is not unique to rectal papillae; the non-sharing brain and optic lobe (*Figure 4—figure supplement 1A*) also express high levels of all three (*Leader et al., 2018*). We examined localization of Inx3 (a gap junction component) (*Curtin et al., 1999; Richard et al., 2017*), and compared it to a septate junction component, NeurexinIV (NrxIV) (*Laprise et al., 2009*). NrxIV localizes similarly both pre and post-sharing onset (*Figure 4D–D'*), indicative of persistent septate junctions remaining between papillar cells. In contrast, Inx3 organizes apically only after cytoplasm sharing (*Figure 4E–E', Figure 4—figure supplement 1B–B'*). Inx3 also does not localize to cell–cell boundaries in *shi* RNAi animals (*Figure 4C–C'*). We tested whether innexins are required for cytoplasm sharing. Knocking down these three genes individually causes mild yet significant cytoplasm sharing defects (*Figure 4F*). However, we see larger defects in animals expressing dominant-negative *ogre*[DN] (*Figure 4F–G; Spéder and Brand, 2014*), which contains a N-terminal GFP tag that interferes with channel passage. Also, heterozygous animals containing a ten gene-deficiency spanning *ogre*, *Inx2*, and *Inx7* have more severe defects (*Figure 4F, Df(1) BSC867*). Finally, we tested whether cytoplasm sharing is essential for normal rectal papillar function. Rectal papillae selectively absorb water and ions from the gut lumen for transport back into the hemolymph, and excrete unwanted lumen contents (*Cohen et al., 2020*). One test of papillar function is viability following the challenge of a high-salt diet (*Bretscher and Fox, 2016; Schoenfelder et al., 2014*). However, with our pan-hindgut driver *byn*-Gal4 used for all previous experiments, we noted animal lethality with *shi*, *Rab5*, and *Rab11* knockdown within a few days on control food. We observed melanization and necrosis throughout the hindgut (data not shown) which prevented us from attributing any phenotypes directly to papillar cytoplasm sharing. We therefore identified an alternative driver (*60H12*-Gal4) with rectum-specific expression during pupation and adulthood (*Figure 4—figure supplement 1D–D'*). We used this driver to express *shi*[DN]. These animals display similar sharing defects as we find with *byn*-Gal4 (*Figure 4—figure supplement 1E–E''*). Reassuringly, *60H12-Gal4 > shi*[DN] animals do not show lethality on a control food diet (*Figure 4H*) allowing us to test rectal papillar physiological function on a high-salt diet. Using either pan-hindgut or papillae-specific knockdown of cytoplasm sharing regulators, we find both *shi*[DN] and *ogre*[DN] animals are extremely sensitive to the high-salt diet (mean survival <1 day, *Figure 4H*). These results underscore an important function for gap junction proteins, as well as membrane remodeling by Dynamin/Shibire, in cytoplasm sharing.

## Discussion

### A distinctive mechanism and model of cytoplasm sharing

Our findings identify *Drosophila* rectal papillae as a new and distinctive example of cytoplasm sharing between multiple nuclei in a simple, genetically tractable system. One defining property of papillar cytoplasm sharing is the lack of an easily observable conduit in the lateral membrane through

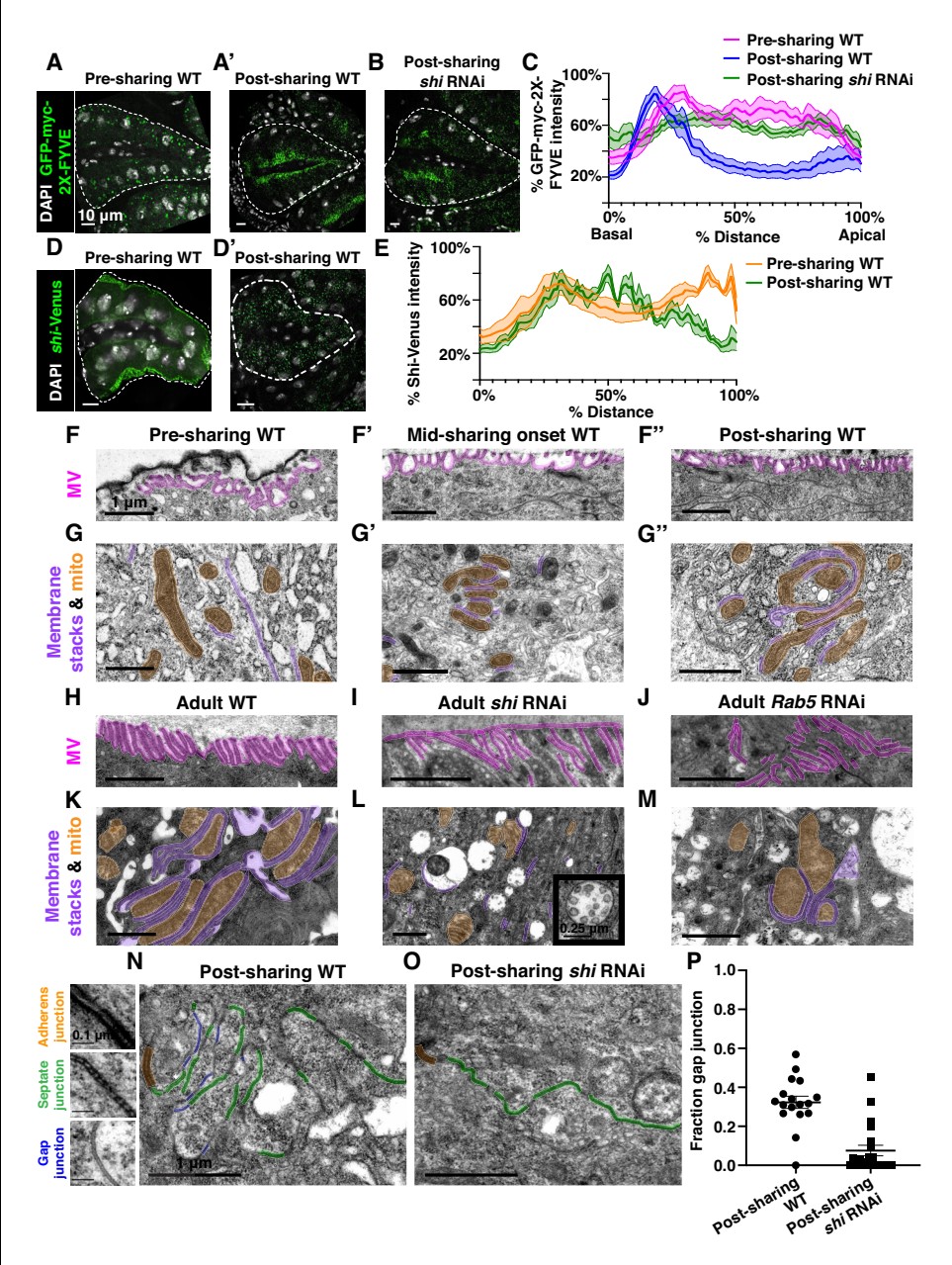

**Figure 3.** Gap junction establishment, but no membrane breaches, accompany cytoplasm sharing. (A–A')
Endosome localization (GFP-myc-2x-FYVE), representative of (A) pre- and (A') post-sharing onset. (B) Endosomes
in *shi RNAi* post-sharing, *see Methods.* (C) Aggregated endosome line profiles for WT pre-sharing (N = 6,
rep = 3), WT post-sharing (N = 7, rep = 2), and *shi RNAi* post-sharing (N = 10, rep = 2). Shaded area represents
standard error. (D–D') Shi-Venus localization pre- and post-sharing onset. (E) Line profiles as in (D–D') (N = 4–5,
rep = 3). (F–O) Representative Transmission Electron Micrographs (TEMs). (F–F'') Microvillar-like structures (MV)
pre- (F), mid- (F'), and post- (F'') sharing onset. (G–G'') Mitochondria and surrounding membrane pre- (G), mid-
(G'), and post- (G'') sharing onset. (H–J) Microvillar-like structures (MV) of adult papillae in WT (H), *shi RNAi* (I), and
*Rab5 RNAi* (J). (K–M) Mitochondria and surrounding membranes of adult papillae in WT (K), *shi RNAi* (L), and *Rab5
RNAi* (M). Inset in (L) shows trapped vesicles. (N–O) WT and *shi RNAi* post-sharing. Adherens (orange), septate
(green), and gap (blue) junctions are highlighted. (P) Quantification of the ratio of gap junction length to septate
plus gap junction length (Fraction gap junction) (N = 3–4, rep = 2). p<0.0001 for the difference in gap junction
ratio between WT and *shi RNAi*.
The online version of this article includes the following figure supplement(s) for figure 3:

*Figure 3 continued*

**Figure supplement 1.** Changes in endosome polarity and apical junction shape accompany the onset of cytoplasm sharing.

**Figure supplement 2.** Extracellular spaces separate nuclei throughout much of the papillar lateral membrane.

which cytoplasm can be exchanged. Cytoplasm sharing in a multinucleate tissue/organism frequently involves the creation of a large membrane breach associated with major actin cytoskeleton rearrangement (*Kim et al., 2015*; *Deng et al., 2017*; *Martin, 2016*). However, papillar cytoplasm sharing does not require canonical myoblast fusion regulators nor major actin remodeling factors such as Rho family GTPases. Aside from membrane breaches, other cell types are known to share cytoplasm through the formation of cytoplasmic bridges such as ring canals or plasmodesmata. Such bridge structures assemble as the result of incomplete cytokinesis (*Mahowald, 1971*; *Lŭcas and Wolf, 1993*). In contrast, papillar cytoplasm sharing does not require mitosis or cytokinesis, and does not contain intercellular bridge structures visible by electron microscopy.

In addition to lacking a large, observable membrane breach, papillar cytoplasm sharing occurs within an intact, polarized epithelium, and apical cell–cell junctions and lateral membranes are retained after the onset of sharing. In contrast, other epithelia known to fuse cytoplasm, such as *C. elegans* epithelia fused by Epithelial Fusion Failure 1 (EFF-1), dismantle cell–cell junctions (*Smurova and Podbilewicz, 2016*). Further, cells with ring canals retain cell–cell junctions and lateral membranes (*Peifer et al., 1993*).

Given the retention of cell junctions and absence of clear intercellular bridges, channels, or breaches in lateral membrane, our data lead us to propose that a specialized function of gap junction proteins facilitates cytoplasm sharing between neighboring cells in an otherwise intact epithelium (*Figure 4I*). Although gap junctions typically transfer molecules of <1 kDa, elongated proteins up to 18 kDa are observed to pass through certain vertebrate gap junctions (*Cieniewicz and Woodruff, 2010*). Alternatively, gap junction-mediated cell to cell communication has been previously implicated in fusion of placental trophoblasts and osteoclasts (*Firth et al., 1980*; *Dunk et al., 2012*; *Schilling et al., 2008*), so we cannot rule out an indirect role for gap junctions in papillar cells, such as through regulation/recruitment of a fusogenic protein (*Petrany and Millay, 2019*). Future work beyond the scope of this study can determine if, for example, papillar gap junctions exhibit a specialized structure to directly facilitate exchange of large cytoplasmic contents. As for the connection between membrane remodeling and gap junction formation, Rab11 has been previously reported to recycle gap junction components in *Drosophila* brain and mammalian cell culture (*Augustin et al., 2017*). Dynamin2 was also implicated in gap junction plaque internalization in mammalian cells (*Gilleron et al., 2011*). However, neither of these factors has been previously implicated in gap junction establishment. We show that Dynamin is required for gap junction formation in papillar cells. Future studies will determine the exact role of Dynamin in gap junction establishment. Another clue for future study is that papillar cytoplasm sharing is developmentally regulated, occurring over a brief 6 hr window, and requires membrane remodeling by trafficking GTPases and gap junction establishment (*Figure 4I*, *Figure 4—figure supplement 1H*). Our results argue that papillar sharing is triggered by a permanent structural rearrangement rather than an active transport mechanism, as the membrane remodelers we identified are required specifically during developmental membrane remodeling.

The mechanisms we report here may be relevant to other emerging roles for membrane remodeling and cytoplasm sharing in the literature. Here, we identify a close relationship between the formation of membrane stacks and cytoplasm sharing. Basolateral membrane infoldings to expand cellular surface area are a common feature of absorptive cells (*Pease, 1956*). The mammalian kidney tubule cells exhibit similar basolateral membrane extensions to which ion transporters such as the Na+/K+-ATPase are localized (*Maunsbach, 1966*; *Molitoris et al., 1992*; *Avner et al., 1992*; *Pease, 1955*; *Sjöstrand and Rhodin, 1953*). Our results suggest that the same membrane remodeling factors that regulate cytoplasm sharing are required for the formation of membrane stacks. To our knowledge, this is the first study to reveal factors involved in basolateral membrane infolding biogenesis. Additionally, our results may also explain other examples of cytoplasm sharing where the underlying mechanism remains to be determined, such as transient cytoplasm sharing in the zebrafish

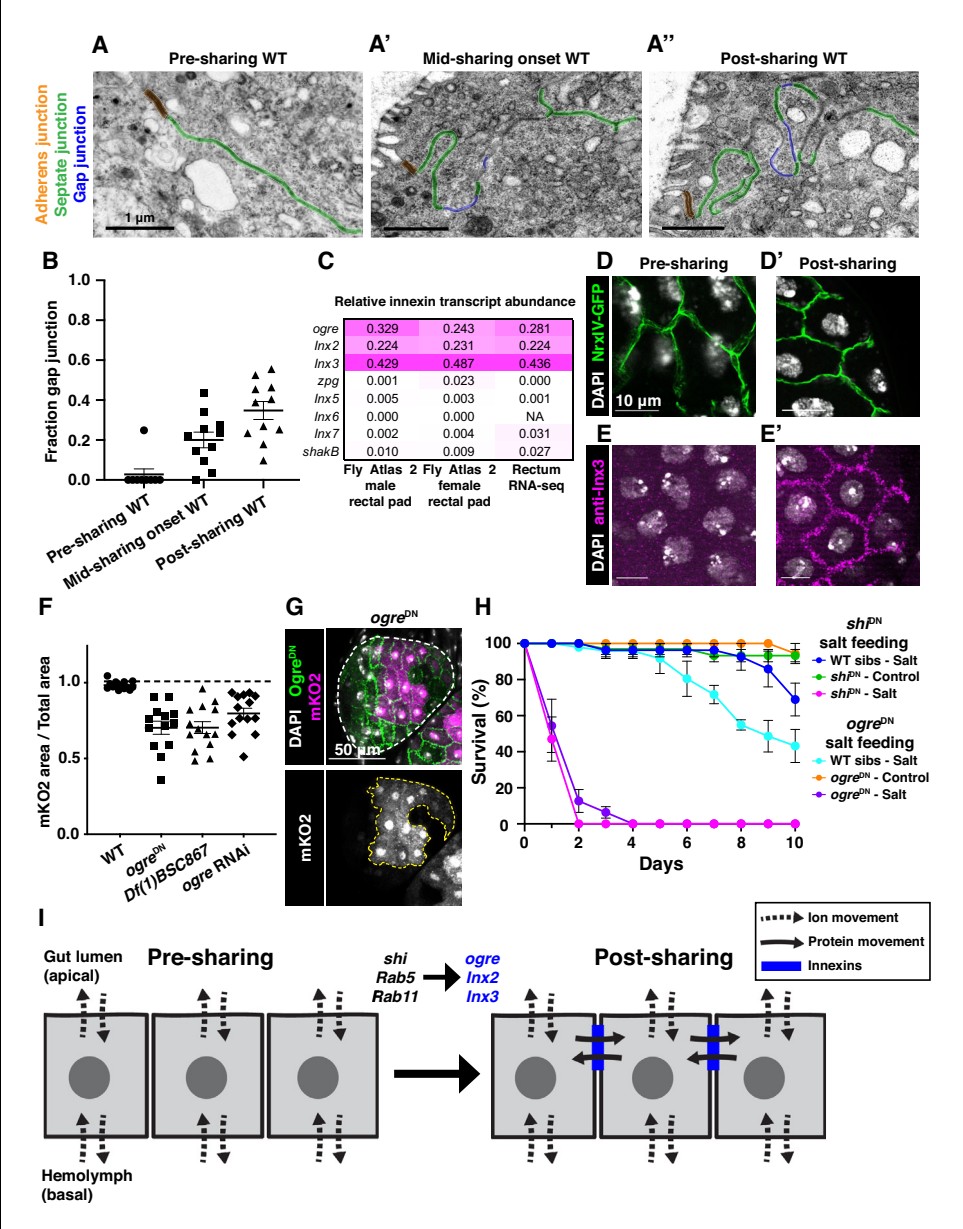

**Figure 4.** Gap junction proteins are required for cytoplasmic sharing. (A–A'') Representative apical junctions highlighted by junctional type in pre (A), mid (A'), and post (A'') sharing onset. (B) Quantification of fraction gap junction (gap junction length / (gap + septate junction length)) in pre-, mid-, and post-sharing onset pupae (N = 3–4, rep = 2). (C) *Drosophila* innexin expression in the adult rectum (*Methods*). (D–D') Adherens junctions in pre- (D) and post- (D') sharing pupae visualized by *NrxIV-GFP*. (E–E') WT pupae pre- and post-sharing onset stained with anti-Inx3. (F) Quantification of cytoplasm sharing in WT, *ogreDN*, *Df(1)BSC867/+* (a 10-gene-deficiency covering *ogre*, *Inx2*, and *Inx7*), and *ogre* RNAi adult papillae (N = 13–14, rep = 2). (G) Representative adult rectal papilla expressing *GFP-ogre* and *dBrainbow*. (H) Survival of WT, *shiDN*, and *ogreDN* animals on a high-salt diet (N = 27–37, rep = 3). (I) Proposed model for cytoplasmic sharing in an intact papillar epithelium.
The online version of this article includes the following figure supplement(s) for figure 4:

**Figure supplement 1.** Gap junction formation coincides with cytoplasm sharing onset.

myocardium (*Sawamiphak et al., 2017*). Together, our studies indicate that the *Drosophila* papillar epithelium represents a distinctive example of cytoplasmic sharing to generate giant multinucleate cells.

# Functions and implications of transforming a multicellular tissue into a giant multinucleate cytoplasm

Our results have several implications for functions and regulation of multinucleation. Here we show that the membrane and junctional changes associated with cytoplasm sharing are required for normal *Drosophila* rectal papillar function. Papillae in other insects are known to undergo visible movement upon muscle contraction, which may facilitate cytoplasm movement (*Lowne, 1869*). Arthropod papillar structures are subject to peristaltic muscle contractions from an extensive musculature (*Rocco et al., 2017*), which aid in both excretion and movement of papillar contents into the hemolymph (*Habas mantel and Mantel, 1968*). Further, relative to other hindgut regions, the rectum appears to have specialized innervation (*Cohen et al., 2020*) and regulation by the kinin family of neuropeptides, which are hypothesized to provide additional input in to muscle activity in this critical site of reabsorption (*Audsley and Weaver, 2009*; *Lajevardi and Paluzzi, 2020*). We speculate that these muscle contractions aid in vigorous movement of papillar cytoplasm, which includes ions and water taken up from the intestinal lumen. The movement of these papillar contents may facilitate both cytoplasm exchange between papillar cells and the interaction of ions and ion transport machinery with intracellular membrane stacks. This idea is supported by our finding that animals lacking a large common papillar cytoplasm die when fed a high-salt diet.

Given the importance of insect papillae in pathogen biology, the knowledge that this common anatomical structure is a shared cytoplasm can impact both human disease intervention and agricultural pest control. Papillae occur in both primitive insect orders such as Zygentoma and Odonata and also in Lepidopterans, Hymenopterans, and Dipterans, the latter of which exhibit the most prominent and elaborate structures (*Palm, 1949*). Furthermore, electron micrographs of the hindgut of the mosquito, *Aedes aegypti*, and the ant, *Formica nigricans*, show striking ultrastructural similarity to *Drosophila*, and these studies leave open the possibility that multinucleation may be conserved in insect papillae (*Hopkins, 1967*; *Wessing and Eichelberg, 1973*; *Garayoa et al., 1999*). Cytoplasm sharing is a known mechanism that facilitates pathogen spread (*Eugenin et al., 2009*), and papillae are an avenue of entry for numerous pathogens including kinetoplastids and mosquito viruses (*Gu et al., 2010*; *Filosa et al., 2019*). Thus, our findings may impact strategies to prevent diseases such as African sleeping sickness, or to target agricultural pests that threaten agricultural production.

The sharing of cytoplasm also has the potential to neutralize detrimental genomic imbalances between nuclei caused by aneuploidy. Our prior work (*Schoenfelder et al., 2014*) revealed that papillae are highly tolerant of chromosome mis-segregation, and our work here suggests this tolerance may be due in part to neutralization of aneuploidies through cytoplasm sharing. This finding may also be relevant to the study of multinucleate tumors, such as those found in pancreas, bone, and fibrous tissues (*Doane et al., 2015*; *Hasegawa et al., 2017*; *Mancini et al., 2017*), or to conditions of aberrant organelle inheritance (*Asare et al., 2017*). Finally, we note that our study reveals that, even in a well-studied model organism such as *Drosophila*, we still have yet to appreciate the full diversity of tissue organization strategies. Our Brainbow-based approach could be applied to other contexts to identify other tissues with cytoplasm sharing, including those with gap junction-dependent but membrane breach-independent cytoplasm sharing. Collectively, our findings highlight the expanding diversity of multicellular tissue organization strategies.

## Materials and methods

### Key resources table

| Reagent type (species) or resource | Designation | Source or reference | Identifiers | Additional information |
|---|---|---|---|---|
| Strain, strain background (*D. melanogaster*) | $w^{1118}$ | Bloomington *Drosophila* Stock Center | BDSC:3605; FLYB:FBst0003605; RRID:BDSC_3605 | $w^{1118}$ |
| Genetic reagent (*D. melanogaster*) | tub-Gal4 | Bloomington *Drosophila* Stock Center | BDSC:5138; FLYB:FBst0005138; RRID:BDSC_5138 | $y^1$ $w^*$; P{tubP-GAL4} LL7/TM3, $Sb^1$ $Ser^1$ |

*Continued on next page*

*Continued*

| Reagent type (species) or resource | Designation | Source or reference | Identifiers | Additional information |
|---|---|---|---|---|
| Genetic reagent (*D. melanogaster*) | *tub-Gal80^ts* | NA | NA | NA |
| Genetic reagent (*D. melanogaster*) | *UAS-dBrainbow* | Bloomington *Drosophila* Stock Center; (*Hampel et al., 2011*) | BDSC:34513; FLYB:FBst0034513; RRID:BDSC_34513 | $w^{1118}$; P{UAS-Brainbow}attP2 |
| Genetic reagent (*D. melanogaster*) | *UAS-dBrainbow* | Bloomington *Drosophila* Stock Center; (*Hampel et al., 2011*) | BDSC:34514; FLYB:FBst0034514; RRID:BDSC_34514 | $w^{1118}$; P{UAS-Brainbow}attP40 |
| Genetic reagent (*D. melanogaster*) | *Hsp70>cre* | Bloomington *Drosophila* Stock Center | BDSC:851; FLYB:FBst0000851; RRID:BDSC_851 | $y^1 w^{67c23}$ P{Crey}1b; $D^*$/TM3, $Sb^1$ |
| Genetic reagent (*D. melanogaster*) | *UAS-fzr RNAi* | Vienna *Drosophila* Resource Center | VDRC:25550; FLYB:FBst0455950 | $w^{1118}$; P{GD9960}v25550 |
| Genetic reagent (*D. melanogaster*) | *UAS-shi RNAi #1* | Bloomington *Drosophila* Stock Center | BDSC:28513; FLYB:FBst0028513; RRID:BDSC_28513 | $y^1 v^1$; P{TRiP.JF03133}attP2 |
| Genetic reagent (*D. melanogaster*) | *UAS-shi RNAi #2* | Bloomington *Drosophila* Stock Center | BDSC:36921; FLYB:FBst0036921; RRID:BDSC_36921 | $y^1 sc^* v^1 sev^{21}$; P{TRiP.HMS00154}attP2 |
| Genetic reagent (*D. melanogaster*) | *UAS-Rab5 RNAi #1* | Bloomington *Drosophila* Stock Center | BDSC:30518; FLYB:FBst0030518; RRID:BDSC_30518 | $y^1 v^1$; P{TRiP.JF03335}attP2 |
| Genetic reagent (*D. melanogaster*) | *UAS-Rab5 RNAi #2* | Bloomington *Drosophila* Stock Center | BDSC:67877; FLYB:FBst0067877; RRID:BDSC_67877 | $y^1 sc^* v^1 sev^{21}$; P{TRiP.GL01872}attP40 |
| Genetic reagent (*D. melanogaster*) | *UAS-Rab11 RNAi #1* | Bloomington *Drosophila* Stock Center | BDSC:27730; FLYB:FBst0027730; RRID:BDSC_27730 | $y^1 v^1$; P{TRiP.JF02812}attP2 |
| Genetic reagent (*D. melanogaster*) | *UAS-Rab11 RNAi #2* | Vienna *Drosophila* Resource Center | VDRC:22198; FLYB:FBst0454467 | $w^{1118}$; P{GD11761}v22198 |
| Genetic reagent (*D. melanogaster*) | *UAS-SCAR RNAi #1* | Bloomington *Drosophila* Stock Center | BDSC:36121; FLYB:FBst0036121; RRID:BDSC_36121 | $y^1 sc^* v^1 sev^{21}$; P{TRiP.HMS01536}attP40 |
| Genetic reagent (*D. melanogaster*) | *UAS-SCAR RNAi #2* | Bloomington *Drosophila* Stock Center | BDSC:51803; FLYB:FBst0051803; RRID:BDSC_51803 | $y^1 v^1$; P{TRiP.HMC03361}attP40 |
| Genetic reagent (*D. melanogaster*) | *UAS-kirre RNAi* | Vienna *Drosophila* Resource Center | VDRC:27227; FLYB:FBst0456824 | $w^{1118}$; P{GD14476}v27227 |
| Genetic reagent (*D. melanogaster*) | *UAS-sns RNAi* | Vienna *Drosophila* Resource Center | VDRC:877; FLYB:FBst0471238 | $w^{1118}$; P{GD65}v877/TM3 |
| Genetic reagent (*D. melanogaster*) | *UAS-schizo RNAi* | Vienna *Drosophila* Resource Center | VDRC:36625; FLYB:FBst0461775 | $w^{1118}$; P{GD14895}v36625 |
| Genetic reagent (*D. melanogaster*) | *UAS-sing RNAi* | Vienna *Drosophila* Resource Center | VDRC:12202; FLYB:FBst0450437 | $w^{1118}$; P{GD3396}v12202/TM3 |
| Genetic reagent (*D. melanogaster*) | *UAS-Cdc42^DN* | Bloomington *Drosophila* Stock Center | BDSC:6288; FLYB:FBst0006288; RRID:BDSC_6288 | $w^*$; P{UAS-Cdc42.N17}3 |

*Continued on next page*

*Continued*

| Reagent type (species) or resource | Designation | Source or reference | Identifiers | Additional information |
|---|---|---|---|---|
| Genetic reagent (*D. melanogaster*) | *UAS-Rac1*$^{DN}$ | Bloomington *Drosophila* Stock Center | BDSC:6292; FLYB:FBst0006292; RRID:BDSC_6292 | y$^1$ w$^*$; P{UAS-Rac1.N17}1 |
| Genetic reagent (*D. melanogaster*) | *UAS-Rho1*$^{DN}$ | Bloomington *Drosophila* Stock Center | BDSC:7328; FLYB:FBst0007328; RRID:BDSC_7328 | w$^*$; P{UAS-Rho1.N19}2.1 |
| Genetic reagent (*D. melanogaster*) | *UAS-GFP*$^{NLS}$ | Bloomington *Drosophila* Stock Center | BDSC:4776; FLYB:FBst0004776; RRID:BDSC_4776 | w$^{1118}$; P{UAS-GFP.nls}8 |
| Genetic reagent (*D. melanogaster*) | *UAS-GFP-Myc-2x-FYVE* | Bloomington *Drosophila* Stock Center | BDSC:42712; FLYB:FBst0042712; RRID:BDSC_42712 | w$^*$; P{UAS-GFP-myc-2xFYVE}2 |
| Genetic reagent (*D. melanogaster*) | *UAS-YFP-Rab5* | Bloomington *Drosophila* Stock Center | BDSC:9775; FLYB:FBst0009775; RRID:BDSC_9775 | y$^1$ w$^*$; P{UASp-YFP.Rab5}Pde8$^{08b}$ |
| Genetic reagent (*D. melanogaster*) | *60H12-Gal4* | Bloomington *Drosophila* Stock Center | BDSC:39268; FLYB:FBst0039268; RRID:BDSC_39268 | w$^{1118}$; P{GMR60H12-GAL4}attP2 |
| Genetic reagent (*D. melanogaster*) | *UAS-shi*$^{DN}$ | Bloomington *Drosophila* Stock Center | BDSC:5822; FLYB:FBst0005822; RRID:BDSC_5822 | w$^*$; TM3, P{UAS-shi.K44A}3-10/TM6B, Tb$^1$ |
| Genetic reagent (*D. melanogaster*) | *NrxIV-GFP* | Bloomington *Drosophila* Stock Center | BDSC:50798; FLYB:FBst0050798; RRID:BDSC_50798 | y$^1$ w$^*$; P{PTT-GA}Nrx-IV$^{CA06597}$ |
| Genetic reagent (*D. melanogaster*) | *Df(1)BSC867* | Bloomington *Drosophila* Stock Center | BDSC:29990; FLYB:FBst0029990; RRID:BDSC_29990 | Df(1)BSC867, w$^{1118}$/Binsinscy |
| Genetic reagent (*D. melanogaster*) | *UAS-ogre RNAi* | Vienna *Drosophila* Resource Center | VDRC:7136; FLYB:FBst0470569 | w$^{1118}$; P{GD3264}v7136 |
| Genetic reagent (*D. melanogaster*) | *byn-Gal4* | ***Singer et al., 1996*** | FLYB:FBal0137290 | P{GawB}byn$^{Gal4}$ |
| Genetic reagent (*D. melanogaster*) | *UAS-GFP*$^{PA}$ | Lynn Cooley; ***McLean and Cooley, 2013*** | FLYB:FBti0148163 | P{20XUAS-IVS-Syn21-mC3PA-GFP-p10} |
| Genetic reagent (*D. melanogaster*) | *UAS-N*$^{DN}$ | ***Rebay et al., 1993*** | NA | NA |
| Genetic reagent (*D. melanogaster*) | *UAS-shi-Venus* | Stefano De Renzis; ***Fabrowski et al., 2013*** | NA | NA |
| Genetic reagent (*D. melanogaster*) | *UAS-GFP-ogre* | Andrea Brand; ***Spéder and Brand, 2014*** | FLYB:FBtp0127574 | ogre$^{UAS.N.GFP}$ |
| Genetic reagent (*D. melanogaster*) | *UAS-Gapdh2-GFP*$^{PA}$ | This paper | NA | Transgenic line created through gene synthesis and embryo injection. Codon-optimized *D. melanogaster* Gapdh2 fused to GFP$^{PA}$ under UAS control. |
| Antibody | anti-GFP (Rabbit polyclonal) | Thermo Fisher Scientific | Cat# A11122; RRID:AB_221569 | IF (1:1000) |
| Antibody | anti-HA (Rat monoclonal) | Roche | Cat# 11867423001; RRID:AB_390918 | IF (1:100) |

*Continued on next page*

*Continued*

| Reagent type (species) or resource | Designation | Source or reference | Identifiers | Additional information |
|---|---|---|---|---|
| Antibody | anti-Inx3 (Rabbit polyclonal) | Reinhard Bauer; *Lehmann et al., 2006* | RRID:AB_2568555 | IF (1:75) |
| Antibody | Anti-Rabbit Alexa Fluor 488 (Goat) | Thermo Fisher Scientific | Cat# A32731; RRID:AB_2633280 | IF (1:2000) |
| Antibody | Anti-Rabbit Alexa Fluor 568 (Goat) | Thermo Fisher Scientific | Cat# A-11011; RRID:AB_143157 | IF (1:2000) |
| Antibody | Anti-Rat Alexa Fluor 633 (Goat) | Thermo Fisher Scientific | Cat# A-21094; RRID:AB_2535749 | IF (1:2000) |
| Other | DAPI stain | Sigma-Aldrich | Cat# D9542 | (1:5000) |

## Fly stocks and genetics

Flies were raised at 25°C on standard media (Archon Scientific, Durham, NC) unless specified otherwise. See *Table 4* for a list of fly stocks used. See *Table 3* for a full list of fly lines screened in primary and secondary screens. See *Table 5* for panel-specific genotypes.

The *UAS-Gapdh2-GFP*$^{PA}$ construct was generated by gene synthesis (Twist Biosciences). The GFP was placed at the C-terminus with a 12-amino acid fusion linker (GSAGSAAGSGEF) (*Waldo et al., 1999*) codon-optimized for *Drosophila*. This insert was then cloned into the pBID-UASC-FG vector modified to remove the FLAG tag and extraneous cloning sites. Transgenic flies were generated at Duke University. *brachyenteron (byn)-Gal4* was the driver for all UAS transgenes with the exception of the screen in *Figure 1—figure supplement 1A*, which used *tub-Gal4*, and the *shi* knockdown in *Figure 4H*, which used *60H12-Gal4*. *60H12-Gal4* expresses only in the papillar cells and not the rest of the hindgut, and use of this driver blocks cytoplasm sharing using *UAS-shi*$^{DN}$ (*Figure 4—figure supplement 1D–E''*). For all *Gal4* experiments, *UAS* expression was at 29°C, except in *Figure 1F–H*, where it was at 25°C. If *byn-Gal4* expression of a given *UAS*-transgene was lethal, the experiment was repeated with a temperature-sensitive *Gal80*$^{ts}$ repressor transgene and animals were kept at 18°C until shifting to 29°C at an experimentally-determined time point that would both result in viable animals and permit time to express the transgene prior to sharing onset.

For salt feeding assays, age- and sex-matched siblings were transferred into vials containing 2% NaCl food made with Nutri-Fly MF food base (Genesee Scientific) or control food (*Schoenfelder et al., 2014*). Flies were monitored for survival each day for 10 days.

## Tissue preparation

For fixed imaging, tissues were dissected in PBS and immediately fixed in 3.7% formaldehyde + 0.3% Triton-X for 15 min. Immunostaining was performed in 0.3% Triton-X with 1% normal goat serum (*Fox et al., 2010*). The following antibodies were used: Rabbit anti-GFP (Thermo Fisher Scientific, Cat#A11122, 1:1000), Rat anti-HA (Roche, Cat#11867423001, 1:100), Rabbit anti-Inx3 (generous gift from Reinhard Bauer, 1:75), [*Lehmann et al., 2006*], 488, 568, 633 secondary antibodies (Thermo Fisher Scientific, Alexa Fluor, 1:2000). Tissue was stained with DAPI at 5 µg/ml and mounted in VECTASHIELD Mounting Media on slides.

## Microscopy

### Light microscopy

For fixed imaging, images were obtained on either a Leica SP5 inverted confocal with a 40X/1.25NA oil objective with emission from a 405 nm diode laser, a 488 nm argon laser, a 561 nm Diode laser, and a 633 HeNe laser under control of Leica LAS AF 2.6 software, or on an Andor Dragonfly Spinning Disk Confocal plus. Images were taken with two different cameras, iXon Life 888 1024 × 1024 EMCCD (pixel size 13 um) and the Andor Zyla PLUS 4.2 Megapixel sCMOS 2048 x 2048 (pixel size 6.5 um) depending on imaging needs. Images were taken on the **40x**/1.25–0.75 oil 11506250: 40X,

**Table 3.** Primary and secondary candidate screen stock numbers used and results.

| Gene | Annotation symbol | Gene ID | Mutant or UAS transgene | Stock center | Stock number | Chr | Sharing disrupted? | Notes |
|---|---|---|---|---|---|---|---|---|
| Abi | CG9749 | FBgn0020510 | RNAi | BDSC | 51455 | 2 | No | |
| ALiX | CG12876 | FBgn0086346 | RNAi | BDSC | 33417 | 3 | No | |
| ALiX | CG12876 | FBgn0086346 | RNAi | BDSC | 50904 | 2 | No | |
| Arf51F | CG8156 | FBgn0013750 | RNAi | BDSC | 51417 | 3 | No | |
| Arf51F | CG8156 | FBgn0013750 | Mutant | BDSC | 17076 | 2 | No | |
| Arf51F | CG8156 | FBgn0013750 | RNAi | BDSC | 27261 | 3 | No | |
| Arp2 | CG9901 | FBgn0011742 | RNAi | BDSC | 27705 | 3 | No | |
| Arp3 | CG7558 | FBgn0262716 | RNAi | BDSC | 32921 | 3 | No | |
| Atg1 | CG10967 | FBgn0260945 | RNAi | BDSC | 44034 | 2 | No | |
| Atg1 | CG10967 | FBgn0260945 | RNAi | BDSC | 26731 | 3 | No | |
| Atg7 | CG5489 | FBgn0034366 | RNAi | BDSC | 34369 | 3 | No | |
| Atg7 | CG5489 | FBgn0034366 | RNAi | BDSC | 27707 | 3 | No | |
| Atg8a | CG32672 | FBgn0052672 | RNAi | BDSC | 28989 | 3 | No | |
| Atg8a | CG32672 | FBgn0052672 | RNAi | BDSC | 58309 | 2 | No | |
| Atg8a | CG32672 | FBgn0052672 | RNAi | BDSC | 34340 | 3 | No | |
| Atl | CG6668 | FBgn0039213 | RNAi | BDSC | 36736 | 2 | No | |
| Bet1 | CG14084 | FBgn0260857 | RNAi | BDSC | 41927 | 2 | No | |
| blue | NA | FBgn0283709 | RNAi | BDSC | 44094 | 3 | No | |
| blue | NA | FBgn0283709 | RNAi | BDSC | 41637 | 2 | No | |
| CadN | CG7100 | FBgn0015609 | RNAi | BDSC | 27503 | 3 | No | |
| CadN | CG7100 | FBgn0015609 | RNAi | BDSC | 41982 | 3 | No | |
| CapD2 | CG1911 | FBgn0039680 | Mutant | BDSC | 59393 | 3 | No | |
| Cdc2 | CG5363 | FBgn0004106 | RNAi | VDRC | 41838 | 3 | Yes | |
| Cdc2 | CG5363 | FBgn0004106 | RNAi | BDSC | NA | 3 | No | |
| Cdc42 | CG12530 | FBgn0010341 | RNAi | BDSC | 42861 | 2 | No | |
| Cdc42 | CG12530 | FBgn0010341 | DN | BDSC | 6288 | 2 | No | |
| Ced-12 | CG5336 | FBgn0032409 | RNAi | BDSC | 28556 | 3 | No | |
| Ced-12 | CG5336 | FBgn0032409 | RNAi | BDSC | 58153 | 2 | No | |
| Chc | CG9012 | FBgn0000319 | DN | BDSC | 26821 | 2 | No | |
| Chc | CG9012 | FBgn0000319 | RNAi | BDSC | 27350 | 3 | No | |
| Chc | CG9012 | FBgn0000319 | RNAi | BDSC | 34742 | 3 | No | |
| Chico | CG5686 | FBgn0024248 | RNAi | BDSC | 36788 | 2 | No | |
| Chmp1 | CG4108 | FBgn0036805 | RNAi | BDSC | 33928 | 3 | No | |
| CHMP2B | CG4618 | FBgn0035589 | RNAi | BDSC | 28531 | 3 | No | |
| CHMP2B | CG4618 | FBgn0035589 | RNAi | BDSC | 38375 | 2 | No | |
| cindr | CG31012 | FBgn0027598 | RNAi | BDSC | 35670 | 3 | No | |
| cindr | CG31012 | FBgn0027598 | RNAi | BDSC | 38976 | 2 | No | |
| Clamp | CG1832 | FBgn0032979 | RNAi | BDSC | 27080 | 3 | No | |
| cno | CG42312 | FBgn0259212 | RNAi | BDSC | 33367 | 3 | No | |
| cno | CG42312 | FBgn0259212 | RNAi | BDSC | 38194 | 2 | No | |
| dac | CG4952 | FBgn0005677 | RNAi | BDSC | 26758 | 3 | No | |
| dac | CG4952 | FBgn0005677 | RNAi | BDSC | 35022 | 3 | No | |
| DCTN1-p150 | CG9206 | FBgn0001108 | DN | BDSC | 51645 | 2 | No | |

*Table 3 continued on next page*

Table 3 continued

| Gene | Annotation symbol | Gene ID | Mutant or UAS transgene | Stock center | Stock number | Chr | Sharing disrupted? | Notes |
|---|---|---|---|---|---|---|---|---|
| *dnd* | CG6560 | FBgn0038916 | RNAi | BDSC | 27488 | 3 | No | |
| *dnd* | CG6560 | FBgn0038916 | RNAi | BDSC | 34383 | 3 | No | |
| *dock* | CG3727 | FBgn0010583 | RNAi | BDSC | 27728 | 3 | No | |
| *dock* | CG3727 | FBgn0010583 | RNAi | BDSC | 43176 | 3 | No | |
| *dock* | CG3727 | FBgn0010583 | Mutant | BDSC | 11385 | 2 | No | |
| *Dr* | CG1897 | FBgn0000492 | RNAi | BDSC | 26224 | 3 | No | |
| *Dr* | CG1897 | FBgn0000492 | RNAi | BDSC | 42891 | 2 | No | |
| *Egfr* | CG10079 | FBgn0003731 | DN | BDSC | 5364 | 2 | Yes | |
| *Egfr* | CG10079 | FBgn0003731 | RNAi | VDRC | 43267 | 3 | Yes | |
| *endos* | CG6513 | FBgn0061515 | RNAi | BDSC | 53250 | 3 | No | |
| *endos* | CG6513 | FBgn0061515 | RNAi | BDSC | 65996 | 3 | No | |
| *Exo70* | CG7127 | FBgn0266667 | RNAi | BDSC | 28041 | 3 | No | |
| *Exo70* | CG7127 | FBgn0266667 | RNAi | BDSC | 55234 | 3 | No | |
| *Exo84* | CG6095 | FBgn0266668 | RNAi | BDSC | 28712 | 3 | Yes | |
| *Flo1* | CG8200 | FBgn0024754 | RNAi | BDSC | 36700 | 3 | No | |
| *Flo1* | CG8200 | FBgn0024754 | RNAi | BDSC | 36649 | 2 | No | |
| *Flo2* | CG32593 | FBgn0264078 | RNAi | BDSC | 55212 | 3 | No | |
| *Flo2* | CG32593 | FBgn0264078 | RNAi | BDSC | 40833 | 2 | No | |
| *fzr* | CG3000 | FBgn0262699 | RNAi | VDRC | 25550 | 2 | Yes | |
| *Gli* | CG3903 | FBgn0001987 | RNAi | BDSC | 31869 | 3 | No | |
| *Gli* | CG3903 | FBgn0001987 | RNAi | BDSC | 58115 | 2 | No | |
| *grk* | CG17610 | FBgn0001137 | RNAi | BDSC | 38913 | 3 | No | |
| *hbs* | CG7449 | FBgn0029082 | RNAi | BDSC | 57003 | 2 | No | |
| *Hem* | CG5837 | FBgn0011771 | Mutant | BDSC | 8752 | 3 | No | |
| *Hem* | CG5837 | FBgn0011771 | Mutant | BDSC | 8753 | 3 | No | |
| *Hem* | CG5837 | FBgn0011771 | RNAi | BDSC | 29406 | 3 | No | |
| *Hem* | CG5837 | FBgn0011771 | RNAi | BDSC | 41688 | 3 | No | |
| *Hsc70Cb* | CG6603 | FBgn0026418 | RNAi | BDSC | 33742 | 3 | No | |
| *Hsc70Cb* | CG6603 | FBgn0026418 | DN | BDSC | 56497 | 2 | No | |
| *Iris* | CG4715 | FBgn0031305 | RNAi | BDSC | 50587 | 2 | No | |
| *Iris* | CG4715 | FBgn0031305 | RNAi | BDSC | 63582 | 2 | No | |
| *l(2)gl* | CG2671 | FBgn0002121 | RNAi | BDSC | 31517 | 3 | No | |
| *lerp* | CG31072 | FBgn0051072 | RNAi | BDSC | 57436 | 2 | No | |
| *lilli* | CG8817 | FBgn0041111 | RNAi | BDSC | 26314 | 3 | No | |
| *lilli* | CG8817 | FBgn0041111 | RNAi | BDSC | 34592 | 3 | No | |
| *mbc* | CG10379 | FBgn0015513 | RNAi | BDSC | 32355 | 3 | No | |
| *mbc* | CG10379 | FBgn0015513 | RNAi | BDSC | 33722 | 3 | No | |
| *Mi-2* | CG8103 | FBgn0262519 | RNAi | BDSC | 16876 | 3 | No | |
| *mtm* | CG9115 | FBgn0025742 | RNAi | BDSC | 38339 | 3 | No | |
| *N* | CG3936 | FBgn0004647 | DN | Rebay Lab | NA | 2 | No | |
| *N* | CG3936 | FBgn0004647 | RNAi | Sara Bray | NA | 1 | No | |
| *Nrg* | CG1634 | FBgn0264975 | RNAi | BDSC | 28724 | 3 | No | |
| *Nrg* | CG1634 | FBgn0264975 | RNAi | BDSC | 38215 | 2 | No | |

*Table 3 continued*

| Gene | Annotation symbol | Gene ID | Mutant or UAS transgene | Stock center | Stock number | Chr | Sharing disrupted? | Notes |
|---|---|---|---|---|---|---|---|---|
| Nrg | CG1634 | FBgn0264975 | RNAi | BDSC | 37496 | 2 | No | |
| nrv3 | CG8663 | FBgn0032946 | RNAi | BDSC | 29431 | 3 | No | |
| nrv3 | CG8663 | FBgn0032946 | RNAi | BDSC | 50725 | 3 | No | |
| nuf | CG33991 | FBgn0013718 | RNAi | BDSC | 31493 | 3 | No | |
| pav | CG1258 | FBgn0011692 | RNAi | BDSC | 35649 | 3 | No | |
| pav | CG1258 | FBgn0011692 | RNAi | BDSC | 43963 | 2 | No | |
| Ptp61F | CG9181 | FBgn0267487 | RNAi | BDSC | 32426 | 3 | No | |
| Ptp61F | CG9181 | FBgn0267487 | RNAi | BDSC | 56036 | 2 | No | |
| Rab1 | CG3320 | FBgn0285937 | CA | BDSC | 9758 | 3 | No | |
| Rab1 | CG3320 | FBgn0285937 | DN | BDSC | 9757 | 3 | Yes | Requires 60H12-Gal4 |
| Rab1 | CG3320 | FBgn0285937 | RNAi | BDSC | 27299 | 3 | Yes | |
| Rab1 | CG3320 | FBgn0285937 | RNAi | BDSC | 34670 | 3 | No | |
| Rab2 | CG3269 | FBgn0014009 | CA | BDSC | 9761 | 2 | No | |
| Rab2 | CG3269 | FBgn0014009 | DN | BDSC | 9759 | 2 | No | |
| Rab3 | CG7576 | FBgn0005586 | CA | BDSC | 9764 | 3 | No | |
| Rab3 | CG7576 | FBgn0005586 | DN | BDSC | 9766 | 2 | No | |
| Rab4 | CG4921 | FBgn0016701 | CA | BDSC | 9770 | 3 | No | |
| Rab4 | CG4921 | FBgn0016701 | DN | BDSC | 9768 | 2 | No | |
| Rab4 | CG4921 | FBgn0016701 | DN | BDSC | 9769 | 3 | No | |
| Rab5 | CG3664 | FBgn0014010 | CA | BDSC | 9773 | 3 | Yes | |
| Rab5 | CG3664 | FBgn0014010 | DN | BDSC | 42704 | 3 | Yes | Requires 60H12-Gal4 |
| Rab5 | CG3664 | FBgn0014010 | RNAi | BDSC | 67877 | 2 | Yes | |
| Rab5 | CG3664 | FBgn0014010 | RNAi | BDSC | 30518 | 3 | Yes | |
| Rab5 | CG3664 | FBgn0014010 | RNAi | BDSC | 51847 | 2 | No | |
| Rab6 | CG6601 | FBgn0015797 | CA | BDSC | 9776 | 3 | No | |
| Rab6 | CG6601 | FBgn0015797 | DN | BDSC | 23250 | 3 | No | |
| Rab7 | CG5915 | FBgn0015795 | CA | BDSC | 9779 | 3 | No | |
| Rab7 | CG5915 | FBgn0015795 | DN | BDSC | 9778 | 3 | No | |
| Rab7 | CG5915 | FBgn0015795 | DN | BDSC | 9778 | 3 | No | |
| Rab8 | CG8287 | FBgn0262518 | DN | BDSC | 9780 | 3 | No | |
| Rab8 | CG8287 | FBgn0262518 | CA | BDSC | 9781 | 2 | No | |
| Rab8 | CG8287 | FBgn0262518 | DN | BDSC | 9780 | 3 | No | |
| Rab9 | CG9994 | FBgn0032782 | CA | BDSC | 9785 | 3 | No | |
| Rab9 | CG9994 | FBgn0032782 | DN | BDSC | 23642 | 3 | No | |
| Rab10 | CG17060 | FBgn0015789 | CA | BDSC | 9787 | 3 | No | |
| Rab10 | CG17060 | FBgn0015789 | DN | BDSC | 9786 | 3 | No | |
| Rab11 | CG5771 | FBgn0015790 | CA | BDSC | 9791 | 3 | No | |
| Rab11 | CG5771 | FBgn0015790 | DN | BDSC | 23261 | 3 | Yes | |
| Rab11 | CG5771 | FBgn0015790 | RNAi | BDSC | 27730 | 3 | Yes | |
| Rab11 | CG5771 | FBgn0015790 | RNAi | VDRC | 108382 | 2 | Yes | |
| Rab11 | CG5771 | FBgn0015790 | RNAi | VDRC | 22198 | 3 | Yes | |
| Rab11 | CG5771 | FBgn0015790 | Mutant | BDSC | 42708 | 3 | Yes | |
| Rab14 | CG4212 | FBgn0015791 | CA | BDSC | 9795 | 2 | No | |

*Table 3 continued on next page*

*Table 3 continued*

| Gene | Annotation symbol | Gene ID | Mutant or UAS transgene | Stock center | Stock number | Chr | Sharing disrupted? | Notes |
|---|---|---|---|---|---|---|---|---|
| *Rab14* | CG4212 | FBgn0015791 | DN | BDSC | 23264 | 3 | No | |
| *Rab18* | CG3129 | FBgn0015794 | CA | BDSC | 9797 | 3 | No | |
| *Rab18* | CG3129 | FBgn0015794 | DN | BDSC | 23238 | 3 | No | |
| *Rab19* | CG7062 | FBgn0015793 | CA | BDSC | 9800 | 3 | No | |
| *Rab19* | CG7062 | FBgn0015793 | DN | BDSC | 9799 | 3 | No | |
| *Rab21* | CG17515 | FBgn0039966 | CA | BDSC | 23864 | 2 | No | |
| *Rab21* | CG17515 | FBgn0039966 | DN | BDSC | 23240 | 3 | No | |
| *Rab23* | CG2108 | FBgn0037364 | RNAi | BDSC | 36091 | 3 | No | |
| *Rab23* | CG2108 | FBgn0037364 | RNAi | BDSC | 55352 | 2 | No | |
| *Rab23* | CG2108 | FBgn0037364 | CA | BDSC | 9806 | 3 | No | |
| *Rab23* | CG2108 | FBgn0037364 | DN | BDSC | 9804 | 3 | No | |
| *Rab26* | CG34410 | FBgn0086913 | CA | BDSC | 23243 | 3 | No | |
| *Rab26* | CG34410 | FBgn0086913 | DN | BDSC | 9808 | 3 | No | |
| *Rab27* | CG14791 | FBgn0025382 | CA | BDSC | 9811 | 2 | No | |
| *Rab27* | CG14791 | FBgn0025382 | DN | BDSC | 23267 | 2 | No | |
| *Rab30* | CG9100 | FBgn0031882 | CA | BDSC | 9814 | 2 | No | |
| *Rab30* | CG9100 | FBgn0031882 | DN | BDSC | 9813 | 3 | No | |
| *Rab32* | CG8024 | FBgn0002567 | CA | BDSC | 23280 | 3 | No | |
| *Rab32* | CG8024 | FBgn0002567 | DN | BDSC | 23281 | 2 | No | |
| *Rab35* | CG9575 | FBgn0031090 | CA | BDSC | 9817 | 3 | No | |
| *Rab35* | CG9575 | FBgn0031090 | DN | BDSC | 9820 | 3 | No | |
| *Rab39* | CG12156 | FBgn0029959 | CA | BDSC | 9823 | 3 | No | |
| *Rab39* | CG12156 | FBgn0029959 | DN | BDSC | 23247 | 3 | No | |
| *Rab40* | CG1900 | FBgn0030391 | CA | BDSC | 9827 | 3 | No | |
| *Rab40* | CG1900 | FBgn0030391 | DN | BDSC | 9829 | 2 | No | |
| *Rab9D* | CG32678 | FBgn0067052 | CA | BDSC | 9835 | 3 | No | |
| *Rab9D* | CG32678 | FBgn0067052 | DN | BDSC | 23257 | 2 | No | |
| *Rab9E* | CG32673 | FBgn0052673 | CA | BDSC | 9832 | 2 | No | |
| *Rab9E* | CG32673 | FBgn0052673 | DN | BDSC | 23255 | 3 | No | |
| *Rab9Fb* | CG32670 | FBgn0052670 | CA | BDSC | 9844 | 3 | No | |
| *Rab9Fb* | CG32670 | FBgn0052670 | DN | BDSC | 9845 | 2 | No | |
| *RabX1* | CG3870 | FBgn0015372 | CA | BDSC | 9839 | 2 | No | |
| *RabX1* | CG3870 | FBgn0015372 | DN | BDSC | 23252 | 3 | No | |
| *RabX2* | CG2885 | FBgn0030200 | CA | BDSC | 9842 | 3 | No | |
| *RabX2* | CG2885 | FBgn0030200 | DN | BDSC | 9843 | 2 | No | |
| *RabX4* | CG31118 | FBgn0051118 | RNAi | BDSC | 28704 | 3 | No | |
| *RabX4* | CG31118 | FBgn0051118 | RNAi | BDSC | 44070 | 2 | No | |
| *RabX4* | CG31118 | FBgn0051118 | CA | BDSC | 23277 | 2 | No | |
| *RabX4* | CG31118 | FBgn0051118 | DN | BDSC | 9849 | 3 | No | |
| *RabX5* | CG7980 | FBgn0035255 | CA | BDSC | 9852 | X | No | |
| *RabX5* | CG7980 | FBgn0035255 | DN | BDSC | 9853 | 2 | No | |
| *RabX6* | CG12015 | FBgn0035155 | CA | BDSC | 9855 | 2 | No | |
| *RabX6* | CG12015 | FBgn0035155 | DN | BDSC | 9856 | 3 | No | |

*Table 3 continued*

| Gene | Annotation symbol | Gene ID | Mutant or UAS transgene | Stock center | Stock number | Chr | Sharing disrupted? | Notes |
|------|------|------|------|------|------|------|------|------|
| CG41099 | CG41099 | FBgn0039955 | RNAi | BDSC | 34883 | 3 | No | |
| Rac1 | CG2248 | FBgn0010333 | RNAi | BDSC | 28985 | 3 | No | |
| Rac1 | CG2248 | FBgn0010333 | DN | BDSC | 6292 | 3 | No | |
| Rala | CG2849 | FBgn0015286 | DN | BDSC | 32094 | 2 | No | |
| Rala | CG2849 | FBgn0015286 | RNAi | BDSC | 34375 | 3 | No | |
| Rbp9 | CG3151 | FBgn0010263 | RNAi | BDSC | 42796 | 3 | No | |
| Rep | CG8432 | FBgn0026378 | RNAi | BDSC | 28047 | 3 | No | |
| rho | CG1004 | FBgn0004635 | Mutant | BDSC | 1471 | 3 | Yes | |
| rho | CG1004 | FBgn0004635 | RNAi | BDSC | 38920 | 3 | Yes | |
| rho | CG1004 | FBgn0004635 | RNAi | BDSC | 41699 | 2 | Yes | |
| Rho1 | CG8416 | FBgn0014020 | DN | BDSC | 7328 | 3 | No | |
| Rho1 | CG8416 | FBgn0014020 | DN | BDSC | 58818 | 2 | No | |
| Rho1 | CG8416 | FBgn0014020 | RNAi | BDSC | 32383 | 3 | No | |
| Rip11 | CG6606 | FBgn0027335 | RNAi | BDSC | 38325 | 3 | No | |
| rols | CG32096 | FBgn0041096 | RNAi | BDSC | 56986 | 2 | No | |
| rols | CG32096 | FBgn0041096 | RNAi | BDSC | 58262 | 2 | No | |
| rst | CG4125 | FBgn0003285 | RNAi | BDSC | 28672 | 3 | No | |
| ru | CG1214 | FBgn0003295 | RNAi | BDSC | 41593 | 3 | No | |
| ru | CG1214 | FBgn0003295 | RNAi | BDSC | 58065 | 2 | No | |
| SA-2 | CG13916 | FBgn0043865 | RNAi | VDRC | 108267 | 2 | No | |
| SCAR | CG4636 | FBgn0041781 | RNAi | BDSC | 31126 | 3 | No | |
| SCAR | CG4636 | FBgn0041781 | RNAi | BDSC | 51803 | 2 | No | |
| SCAR | CG4636 | FBgn0041781 | Mutant | BDSC | 8754 | 2 | No | |
| sdt | CG32717 | FBgn0261873 | RNAi | BDSC | 33909 | 3 | No | |
| sdt | CG32717 | FBgn0261873 | RNAi | BDSC | 35291 | 3 | No | |
| Sec10 | CG6159 | FBgn0266673 | RNAi | BDSC | 27483 | 3 | Yes | |
| Sec15 | CG7034 | FBgn0266674 | RNAi | BDSC | 27499 | 3 | Yes | |
| Sec5 | CG8843 | FBgn0266670 | RNAi | VDRC | 28873 | 3 | Yes | |
| Sec5 | CG8843 | FBgn0266670 | RNAi | BDSC | 50556 | 3 | No | |
| Sec6 | CG5341 | FBgn0266671 | RNAi | VDRC | 105836 | 2 | Yes | |
| Sec6 | CG5341 | FBgn0266671 | RNAi | BDSC | 27314 | 3 | Yes | |
| Sec8 | CG2095 | FBgn0266672 | RNAi | BDSC | 57441 | 2 | Yes | |
| shg | CG3722 | FBgn0003391 | RNAi | BDSC | 27689 | 3 | No | |
| shi | CG18102 | FBgn0003392 | DN | BDSC | 5822 | 3 | Yes | Requires 60H12-Gal4 |
| shi | CG18102 | FBgn0003392 | RNAi | BDSC | 28513 | 3 | Yes | |
| shi | CG18102 | FBgn0003392 | RNAi | BDSC | 36921 | 3 | Yes | |
| siz | CG32434 | FBgn0026179 | RNAi | BDSC | 39060 | 2 | No | |
| spi | CG10334 | FBgn0005672 | RNAi | BDSC | 28387 | 3 | No | |
| spi | CG10334 | FBgn0005672 | RNAi | BDSC | 34645 | 3 | No | |
| stet | CG33166 | FBgn0020248 | RNAi | BDSC | 57698 | 3 | No | |
| Vha16-1 | CG3161 | FBgn0262736 | RNAi | BDSC | 40923 | 2 | Yes | |
| Vha16-1 | CG3161 | FBgn0262736 | RNAi | VDRC | 104490 | 2 | Yes | |
| Vha16-1 | CG3161 | FBgn0262736 | RNAi | VDRC | 49291 | 2 | Yes | |

*Table 3 continued on next page*

*Table 3 continued*

| Gene | Annotation symbol | Gene ID | Mutant or UAS transgene | Stock center | Stock number | Chr | Sharing disrupted? | Notes |
|---|---|---|---|---|---|---|---|---|
| *Vha16-2* | CG32089 | FBgn0028668 | RNAi | BDSC | 65167 | 2 | No | |
| *Vha16-3* | CG32090 | FBgn0028667 | RNAi | BDSC | 57474 | 2 | No | |
| *Vha16-5* | CG6737 | FBgn0032294 | RNAi | BDSC | 25803 | 3 | Yes | |
| *Vha55* | CG17369 | FBgn0005671 | RNAi | BDSC | 40884 | 2 | No | |
| *VhaAC39-1* | CG2934 | FBgn0285910 | RNAi | BDSC | 35029 | 3 | No | |
| *VhaAC39-2* | CG4624 | FBgn0039058 | Mutant | BDSC | 62725 | 3 | No | |
| *VhaAC39-2* | CG4624 | FBgn0039058 | RNAi | VDRC | 34303 | 2 | No | |
| *VhaPPA1-1* | CG7007 | FBgn0028662 | RNAi | BDSC | 57729 | 2 | Yes | |
| *VhaPPA1-2* | CG7026 | FBgn0262514 | RNAi | BDSC | 65217 | 2 | Yes | |
| *Vps2* | CG14542 | FBgn0039402 | RNAi | VDRC | 24869 | 3 | Yes | |
| *Vps2* | CG14542 | FBgn0039402 | RNAi | BDSC | 38995 | 2 | Yes | |
| *lsn* | CG6637 | FBgn0260940 | RNAi | BDSC | 38289 | 2 | No | |
| *Vps29* | CG4764 | FBgn0031310 | RNAi | BDSC | 53951 | 2 | No | |
| *Vps33b* | CG5127 | FBgn0039335 | RNAi | BDSC | 44006 | 2 | No | |
| *Vps35* | CG5625 | FBgn0034708 | RNAi | BDSC | 38944 | 2 | No | |
| *Vps4* | CG6842 | FBgn0283469 | RNAi | BDSC | 31751 | 3 | No | |
| *wts* | CG12072 | FBgn0011739 | RNAi | BDSC | 41899 | 3 | No | |
| *wash* | CG13176 | FBgn0033692 | RNAi | BDSC | 62866 | 2 | No | |
| *WASp* | CG1520 | FBgn0024273 | RNAi | BDSC | 25955 | 3 | No | |
| *WASp* | CG1520 | FBgn0024273 | RNAi | BDSC | 51802 | 2 | No | |
| *βggt-II* | CG18627 | FBgn0028970 | RNAi | BDSC | 50516 | 2 | No | |
| *βggt-II* | CG18627 | FBgn0028970 | RNAi | BDSC | 34902 | 3 | No | |

HCX PL APO, NA: 1.25, Oil, DIC, WD: 0.1 mm, coverglass: 0.17 mm, Iris diaphragm, Thread type: M25, **63x**/1.20 water 11506279: 63X, HCX PL APO W Corr CS, NA: 1.2, Water, DIC, WD: 0.22 mm, Coverglass: 0.14–0.18mm, thread type: M25, and **100x**/1.4–0.70 oil 11506210: HCX PL APO, NA: 1.4, Oil, DIC, WD: 0.09 mm, Coverglass: 0.17 mm, Iris Diaphragm, Thread type: M25. The lasers used were: 405 nm diode laser, 488 nm argon laser, 561 nm diode laser, and HeNe 633 nm laser.

For live imaging, hindguts were dissected and cultured based on previous protocols (*Fox et al., 2010*). Live imaging of cell fusion was performed on a spinning disc confocal (Yokogawa CSU10 scanhead) on an Olympus IX-70 inverted microscope using a 40x/1.3 NA UPlanFl N Oil objective, a 488 nm and 568 nm Kr-Ar laser lines for excitation and an Andor Ixon3 897 512 EMCCD camera. The system was controlled by MetaMorph 7.7.

Photo-activation was carried out using Leica SP5 and SP8 microscopes and the FRAP Wizard embedded in the Leica AS-F program. An initial z-stack of the tissue was acquired both before and after activation to examine the full extent of GFP[PA] movement in three dimensions. GFP[PA] transgenes were activated by either point activation or region of interest activation with the 405 nm laser set to between 5 and 20%, depending on the microscope and sample of interest. For each imaging session, test activations on nearby tissues were performed prior to quantify experiments to ensure that only single cells were being activated. After activation, the wizard software was used to acquire time lapses of 15 s to 2min of a single activation plane in order to capture protein movement. Extremely low 488 nm and 405 nm laser powers were used in acquisition of the time lapse images of GFP and Hoechst respectively. Low level 405 nm scanning did not significantly activate GFP[PA], and control experiments were performed without the use of 405 nm time lapses and showed the same protein movement results (data not shown).

**Table 4.** Fly stocks used in addition to the screens.

| Stock name | Stock number | Origin | References |
|---|---|---|---|
| $w^{1118}$ | 3605 | BDSC | |
| tub-Gal4 | 5138 | BDSC | |
| tub-Gal80$^{ts}$ | NA | NA | |
| UAS-dBrainbow | 34513 | BDSC | *Hampel et al., 2011* |
| UAS-dBrainbow | 34514 | BDSC | *Hampel et al., 2011* |
| Hsp70 > cre | 851 | BDSC | |
| UAS-fzr RNAi | 25550 | VDRC | *Fox et al., 2010; Schoenfelder et al., 2014* |
| UAS-shi RNAi #1 | 28513 | BDSC | |
| UAS-shi RNAi #2 | 36921 | BDSC | |
| UAS-Rab5 RNAi #1 | 30518 | BDSC | |
| UAS-Rab5 RNAi #2 | 67877 | BDSC | |
| UAS-Rab11 RNAi #1 | 27730 | BDSC | |
| UAS-Rab11 RNAi #2 | 22198 | VDRC | |
| UAS-SCAR RNAi #1 | 36121 | BDSC | *Bischoff et al., 2013* |
| UAS-SCAR RNAi #2 | 51803 | BDSC | *Xing et al., 2018* |
| UAS-kirre RNAi | 27227 | VDRC | *Linneweber et al., 2015* |
| UAS-sns RNAi | 877 | VDRC | *Linneweber et al., 2015* |
| UAS-schizo RNAi | 36625 | VDRC | *Johnson et al., 2011* |
| UAS-sing RNAi | 12202 | VDRC | *Brunetti et al., 2015* |
| UAS-Cdc42$^{DN}$ | 6288 | BDSC | |
| UAS-Rac1$^{DN}$ | 6292 | BDSC | |
| UAS-Rho1$^{DN}$ | 7328 | BDSC | |
| UAS-GFP$^{NLS}$ | 4776 | BDSC | |
| UAS-GFP-Myc-2x-FYVE | 42712 | BDSC | *Gillooly et al., 2000; Wucherpfennig et al., 2003* |
| UAS-YFP-Rab5 | 9775 | BDSC | |
| 60H12-Gal4 | 39268 | BDSC | |
| UAS-shi$^{DN}$ | 5822 | BDSC | |
| NrxIV-GFP | 50798 | BDSC | |
| Df(1)BSC867 | 29990 | BDSC | |
| UAS-ogre RNAi | 7136 | VDRC | *Holcroft et al., 2013; Spéder and Brand, 2014* |
| byn-Gal4 | - | NA | *Singer et al., 1996* |
| UAS-GFP$^{PA}$ | - | Lynn Cooley | *Datta et al., 2008* |
| UAS-N$^{DN}$ | - | NA | *Rebay et al., 1993* |
| UAS-shi-Venus | - | Stefano De Renzis | *Fabrowski et al., 2013* |
| UAS-GFP-ogre | - | Andrea Brand | *Spéder and Brand, 2014* |
| UAS-Gapdh2-GFP$^{PA}$ | - | - | This paper |

## Transmission electron microscopy

Hindguts were dissected into PBS and fixed in a solution of 2.5% glutaraldehyde in 0.1% cacodylate buffer, pH 7.2. Post-fix specimens were stained with 1% osmium tetroxide in 0.1M cacodylate buffer, dehydrated, soaked in a 1:1 propylene oxide:Epon 812 resin, and then embedded in molds with fresh Epon 812 resin at 65°C overnight. The blocks were cut into semi-thin (0.5 µm) sections using Leica Reichert Ultracuts and the sections were stained with 1% methylene blue. After inspection,

**Table 5.** Additional Methods.

| Panel | Additional methods |
|---|---|
| *Figure 1—figure supplement 1F-F''* | *Hsp70 > cre; UAS-dBrainbow; byn-Gal4* papillae dissected at 62 (D), 69 (D'), or 80 (D'') hours post-puparium formation (HPPF) at 25°C. Hindguts were stained with Rabbit anti-GFP (Thermo-Fisher, A11122, 1:1000), Rat anti-HA (Sigma, 3F10, 1:100), and DAPI at 5 μg/ml. |
| *Figure 1G* | *Hsp70 > cre; UAS-dBrainbow; byn-Gal4* papillae dissected at various HPPF at 25°C. The area labeled by mKO2 was divided by total papillar area. |
| *Figure 1H* | *Hsp70 > cre; UAS-dBrainbow; byn-Gal4* papillae live-imaged at 69HPPF at 25°C. |
| *Figure 1H'* | Fluorescence intensity measured in neighboring cells during sharing onset (1H). |
| *Figure 1I-I'* | *byn-Gal4/UAS-GFP^PA*, live-imaged during adulthood. Single secondary and principal cells were photoactivated and imaged every 3 s. |
| *Figure 2A* | UAS-RNAis and dominant-negative versions of 77 genes representing a wide range of cellular roles were screened (*Hsp70 > cre; UAS-dBrainbow; byn-Gal4*) for sharing defects. Animals expressing both *UAS-dBrainbow* and an *UAS*-driven RNAi or mutant gene were raised at 25°C and shifted to 29°C at L3. If a given RNAi or DN line was lethal when expressed with the *byn-Gal4* driver, a *Gal80^ts* was crossed in and the animals raised at 18°C with a shift to 29°C at pupation. Given the robustness of cytoplasmic sharing in WT animals, gene knockdowns or mutants with even single cell defects in sharing were considered 'hits'. |
| *Figure 2B* | Secondary screen of 36 genes representing various categories of membrane trafficking (*Hsp70 > cre; UAS-dBrainbow; byn-Gal4*) for sharing defects. Animals expressing both *UAS-dBrainbow* and an *UAS*-driven RNAi were raised at 25°C and shifted to 29°C at L3. If a given RNAi line was lethal when expressed with the *byn-Gal4* driver, a *Gal80^ts* was crossed in and the animals raised at 18°C with a shift to 29°C at pupation. Given the robustness of cytoplasmic sharing in WT animals, gene knockdowns with even single cell defects in sharing were considered 'hits'. |
| *Figure 2C* | Secondary screen (*Hsp70 > cre; UAS-dBrainbow; byn-Gal4*) of dominant-negative and constitutively-active variants of the Drosophila Rab GTPases. *UAS-Rab11^DN* and *UAS-Rab14^DN* required a *Gal80^ts* repressor and temperature shifts from 18 to 29°C at pupation. *UAS-Rab1^DN* and *UAS-Rab5^DN* required papillar-specific expression using an alternative *Gal4* driver (*60*H12-Gal4), *Gal80^ts* repressor, and temperature shifts from 18 to 29°C at pupation. |
| *Figure 2D* | *Hsp70 > cre; UAS-dBrainbow; byn-Gal4, Gal80^ts* animals dissected pre-sharing (48 HPPF at 29°C). |
| *Figure 2D'* | *Hsp70 > cre; UAS-dBrainbow; byn-Gal4, Gal80^ts* animals raised at 18°C and shifted to 29°C at pupation and dissected post-sharing (young adult). |
| *Figure 2E* | Young adult animals expressing *UAS-shi RNAi #1* in a *Hsp70 > cre; UAS-dBrainbow; byn-Gal4, Gal80^ts* background. Animals were shifted from 18 to 29°C at pupation to maximize RNAi and minimize animal lethality. |
| *Figure 2F* | Young adult animals expressing *UAS-Rab5 RNAi #1* in a *Hsp70 > cre; UAS-dBrainbow; byn-Gal4, Gal80^ts* background. Animals were shifted from 18 to 29°C at 1–2 days PPF to maximize RNAi and minimize animal lethality. |
| *Figure 2G* | Young adult animals expressing *UAS-Rab11 RNAi #2* in a *Hsp70 > cre; UAS-dBrainbow; byn-Gal4, Gal80^ts* background. Animals were shifted from 18 to 29°C at 1–2 days PPF to maximize RNAi and minimize animal lethality. |
| *Figure 2H* | Animals were shifted and dissected as in 2D-G. Additionally, *Hsp70 > cre; UAS-dBrainbow; byn-Gal4, Gal80^ts* animals expressing *UAS-shi RNAi #2* were raised at 18°C and shifted to 29°C at pupation, animals expressing *UAS-Rab5 RNAi #2* were raised at 18°C and shifted to 29°C at L3, and animals expressing *UAS-Rab11 RNAi #1* were raised at 18°C and shifted to 29°C at 1–2 days PPF. |
| *Figure 3A-A'* | Pupae expressing the early and late endosome marker *UAS-GFP-myc-2x-FYVE* were dissected pre (A, 48HPPF at 29°C) and post (A', 72HPPF at 29°C) sharing onset. |
| *Figure 3B* | Pupae expressing *UAS-GFP-myc-2x-FYVE* in a *UAS-shi RNAi #1* background at a post-sharing time point (24HPPF at 18°C + 72 hr at 29°C). |
| *Figure 3C* | Aggregated line profiles of *UAS-GFP-myc-2x-FYVE* intensity across papilla. |
| *Figure 3D-D'* | Pupae expressing *UAS-shi-Venus* were dissected pre (D, 48HPPF at 29°C) and post (D', 72HPPF at 29°C) sharing onset. |
| *Figure 3E* | Aggregated line profiles of Shi-Venus intensity from the basal (0% distance) to the apical (100% distance) edges of the papilla. See 3C. |
| *Figure 3F-F''* | Transmission electron micrographs of the microvillar-like structures of pupal papillae pre (F, 60HPPF at 25°C), mid (F', 66HPPF at 25°C), and post (F'', 69HPPF at 25°C) cytoplasm sharing onset. |
| *Figure 3G-G''* | Electron micrographs of mitochondria and surrounding membrane material pre (G, 60HPPF at 25°C), mid (G', 66HPPF at 25°C), and post (G'', 69HPPF at 25°C) |
| *Figure 3H* | Electron micrograph of microvillar-like structures of WT (*w^1118*) young adult papillar cells. |
| *Figure 3I* | Electron micrograph of microvillar-like structures of young adult *byn-Gal4, Gal80^ts, UAS-shi RNAi #2* (raised at 18°C, shifted at pupation to 29°C). |
| *Figure 3J* | Electron micrograph of microvillar-like structures of young adult *byn-Gal4, Gal80^ts, UAS-Rab5 RNAi #1* animals (raised at 18°C, shifted at 1–2 days PPF to 29°C). |
| *Figure 3K* | Electron micrograph of mitochondria and surrounding membrane material of WT (*w^1118*) young adult papillar cells. |
| *Figure 3L* | Electron micrograph of mitochondria and surrounding membrane material of young adult *byn-Gal4, Gal80^ts, UAS-shi RNAi #2* (raised at 18°C, shifted at pupation to 29°C). |

*Table 5 continued on next page*

Table 5 continued

| Panel | Additional methods |
|---|---|
| *Figure 3M* | Electron micrograph of mitochondria and surrounding membrane material of young adult *byn-Gal4, Gal80^{ts}, UAS-Rab5 RNAi #1* animals (raised at 18°C, shifted at 1–2 days PPF to 29°C). |
| *Figure 3N* | Electron micrograph of post-sharing WT (TM3/*UAS-shi RNAi #1*) pupa (24HPPF at 18°C, shifted to 29°C for 50 hr, then dissected) |
| *Figure 3O* | Electron micrograph of post-sharing *byn-Gal4, Gal80^{ts},UAS-shi RNAi #1* pupa (24HPPF at 18°C, shifted to 29°C for 50 hr, then dissected) |
| *Figure 3P* | Gap junction length / (gap junction length + septate junction length) measured in WT and *UAS-shi RNAi #1* pupae (see 3N-3O). Each point represents an image of a junction. |
| *Figure 4A-A''* | Electron micrographs of apical junctions (adherens, septate, and gap) pre (A, 60HPPF at 25°C), mid (A', 66HPPF at 25°C), and post (A'', 69HPPF at 25°C) |
| *Figure 4B* | Gap junction length / (gap junction length + septate junction length) measured in pupae pre (60HPPF at 25°C), mid (66HPPF at 25°C), and post (69HPPF at 25°C) sharing onset. Each point represents an image of a junction. |
| *Figure 4C* | Relative innexin transcript abundance (innexin X transcripts/total innexin transcripts) using data from Fly Atlas 2 (*Leader et al., 2018*) and RNA-seq of adult *w^{1118}* rectums performed in the Fox Lab. |
| *Figure 4D-D'* | Pupae with endogenously GFP-tagged NrxIV (*NrxIV-GFP*) dissected pre (D, 48HPPF) and post (D', 72HPPF) sharing onset. |
| *Figure 4E-E'* | Pupae stained with Inx3 antibody (gift from Reinhard Bauer, rabbit, 1:75) pre (E, 48HPPF) and post (E', 58HPPF, papillae do not stain well at later timepoints) sharing onset. |
| *Figure 4F* | Young adult animals expressing no transgene (WT), *UAS-ogre^{DN}*, *UAS-ogre RNAi*, or containing a deficiency covering *ogre*, *Inx2*, and *Inx7* in a *Hsp70 > cre; UAS-dBrainbow; byn-Gal4, Gal80^{ts}* background. Animals were raised at 25°C until L3 and then shifted to 29°C until dissection at young adulthood. |
| *Figure 4G* | See *Figure 4F*. |
| *Figure 4H* | *60 H12-Gal4, Gal80^{ts}* driving *UAS-shi^{DN}* and WT siblings were shifted from 18 to 29°C at pupation. *byn-Gal4, Gal80^{ts}* driving *UAS-ogre^{DN}* animals and WT siblings were raised at 25°C and shifted to 29°C at L3. Animals 1–3 days post-eclosion were sorted into sex-matched groups and fed a control diet or a high salt (2% NaCl) diet. Survival was assessed once per day for 10 days. |
| *Figure 1— figure supplement 1A* | *Hsp70 > cre; UAS-dBrainbow; tubulin-Gal4* animals raised at 29°C. Tissues dissected at adulthood. |
| *Figure 1—figure supplement 1D* | *byn-Gal4/UAS-Gapdh2-GFP^{PA}* raised at 29°C and live-imaged during adulthood. Principal cells were photoactivated and imaged every 15 s. |
| *Figure 1—figure supplement 1E* | *Hsp70 > cre*; UAS-dBrainbow; *byn-Gal4* animals were shifted from 25 to 29°C during L3 and dissected at adulthood. |
| *Figure 1—figure supplement 1F* | *Hsp70 > cre*; UAS-dBrainbow/*UAS-fzr RNAi*; *byn-Gal4* animals were shifted from 25 to 29°C during L2 to maximize *fzr* knock down during endocycling. Animals were dissected at adulthood. |
| *Figure 1—figure supplement 1G* | *Hsp70 > cre*; UAS-dBrainbow; *byn-Gal4/UAS-N^{DN}* animals were shifted from 25 to 29°C during L3 to ensure maximum *UAS-N^{DN}* expression during mitoses. Animals were dissected at adulthood. |
| *Figure 2—figure supplement 1A* | *Hsp70 > cre*; UAS-dBrainbow; *byn-Gal4, Gal80^{ts}* animals expressing various previously published myoblast fusion RNAis raised at 25°C and shifted to 29°C at L3 and dissected post-sharing (young adult). |
| *Figure 2—figure supplement 1B* | *Hsp70 > cre*; UAS-dBrainbow; *byn-Gal4, Gal80^{ts}* animals expressing various previously published UAS-dominant-negative active regulators raised at 18°C and shifted to 29°C at L3 and dissected post-sharing (young adult). |
| *Figure 2—figure supplement 1C* | Papillar cells were identified using *byn-Gal4, Gal80^{ts}*, driving *UAS-GFP^{NLS}* expression. Cells were counted in one, z-sectioned half of the papillae and multiplied by two to give an approximate cell count. |
| *Figure 2—figure supplement 1D* | *Hsp70 > cre*; UAS-dBrainbow; *byn-Gal4, Gal80^{ts}* animals were raised at 18°C until 3–4 days PPF and shifted to 29°C and dissected at young adulthood. |
| *Figure 2—figure supplement 1E* | *Hsp70 > cre*; UAS-dBrainbow; *byn-Gal4, Gal80^{ts}* animals expressing *UAS-shi RNAi #1* were raised at 18°C until 3–4 days PPF and shifted to 29°C and dissected at young adulthood. |
| *Figure 3—figure supplement 1A* | See *Figure 3A-C*. Basal and apical membrane defined as 10–20% and 90–100% total distance of papillae, respectively. |
| *Figure 3—figure supplement 1B-B'* | *byn-Gal4 > UAS-Rab5-YFP* animals dissected pre (48HPPF, 29°C) and post (72HPPF, 29°C) sharing onset. |
| *Figure 3—figure supplement 1B''* | See *Figure 3—figure supplement 1B-B'* and *Figure 3C*. |
| *Figure 3—figure supplement 1C-C''* | Electron micrographs of apical junctions (adherens, septate, and gap) pre (D, 60HPPF at 25°C), mid (D', 66HPPF at 25°C), and post (D'', 69HPPF at 25°C) |
| *Figure 3—figure supplement 1D* | Electron micrograph of apical junctions (adherens, septate, and gap) of WT (*w^{1118}*) young adult papillar cells. |

*Table 5 continued on next page*

*Table 5 continued*

| Panel | Additional methods |
|---|---|
| *Figure 3—figure supplement 1E* | Electron micrograph of apical junctions (adherens, septate, and gap) of young adult *byn-Gal4, Gal80ᵗˢ*ts, *UAS-shi RNAi #2* (raised at 18˚C, shifted at pupation to 29˚C). |
| *Figure 3—figure supplement 1F* | Electron micrograph of apical junctions (adherens, septate, and gap) of young adult *byn-Gal4, Gal80ᵗˢ*ts, *UAS-Rab5 RNAi #1* animals (raised at 18˚C, shifted at 1–2 days PPF to 29˚C). |
| *Figure 3—figure supplement 1G* | See *Figure 3N-O*. Junction width was measured throughout and averaged per image. Each point represents one image of a junction. |
| *Figure 3—figure supplement 1G'* | See *Figure 3N-O*. Junction width was measured throughout and averaged per image. Each point represents one image of a junction. |
| *Figure 3—figure supplement 1G''* | See *Figure 3N-O*. Raw lengths shown were used to calculate 'fraction gap junction' in 3P. Each point represent one image of a junction. |
| *Figure 3—figure supplement 2A* | TEM of young adult (*w¹¹¹⁸*) papilla. |
| *Figure 4—figure supplement 1A* | See *Figure 4A-B*. Junction width was measured throughout and averaged per image. Each point represents one image of a junction. |
| *Figure 4—figure supplement 1A'* | See *Figure 4A-B*. Junction width was measured throughout and averaged per image. Each point represents one image of a junction. |
| *Figure 4—figure supplement 1A''* | See *Figure 4A-B*. Raw lengths shown were used to calculate 'fraction gap junction' in 3P. Each point represent one image of a junction. |
| *Figure 4—figure supplement 1B-B'* | Pupae expressing *byn-Gal4, Gal80ᵗˢ*ts, *UAS-ogreᴰᴺ* (*UAS-GFP-ogre*) dissected pre (B, 48HPPF, 29˚C) and post (B', 72HPPF, 29˚C) sharing onset. |
| *Figure 4—figure supplement 1C* | *byn-Gal4, Gal80ᵗˢ* pupae raised at 18˚C until 0HPPF and then shifted to 29˚C until dissection at 58HPPF. Pupal rectums were stained with Inx3 antibody (gift from Reinhard Bauer, rabbit, 1:75). |
| *Figure 4—figure supplement 1C'* | *byn-Gal4, Gal80ᵗˢ*ts, *UAS-shi RNAi #2* pupae raised at 18˚C until 0HPPF and then shifted to 29˚C until dissection at 58HPPF. Pupal rectums were stained with Inx3 antibody (gift from Reinhard Bauer, rabbit, 1:75). |
| *Figure 4—figure supplement 1D* | *byn-Gal4 > UAS-GFPᴺᴸˢ* dissected pre (48HPPF, 29˚C) sharing onset. |
| *Figure 4—figure supplement 1D'* | *60H12-Gal4 > UAS-GFPᴺᴸˢ* dissected pre (48HPPF, 29˚C) sharing onset. The pan-hindgut driver used in previous experiments, *brachyenteron* (*byn-Gal4*), causes animal lethality with *shi*, *Rab5*, and *Rab11* knockdown within a few days. We therefore screened for and identified an alternative, papillae-specific driver (*60H12-Gal4*), derived from regulatory sequences of the hormone receptor gene *Proctolin Receptor*. *60H12-Gal4 > shiᴰᴺ* animals are viable on a control diet allowing us to test papillar function on a high-salt diet. |
| *Figure 4—figure supplement 1E* | *Hsp70 > cre; UAS-dBrainbow; 60H12-Gal4* animals raised at 18˚C and shifted to 29˚C at pupation and dissected as young adults. |
| *Figure 4—figure supplement 1E'* | *Hsp70 > cre; UAS-dBrainbow; 60H12-Gal4 / UAS-shiᴰᴺ* animals raised at 18˚C and shifted to 29˚C at pupation and dissected as young adults. |
| *Figure 4—figure supplement 1E''* | See *Figure 4—figure supplement 1E-E'*. |

ultra-thin sections (65−75 nm) were cut using Leica EM CU7 and contrast stained with 2% uranyl acetate, 3.5% lead citrate solution. Ultrathin sections were visualized on a JEM-1400 transmission electron microscope (JEOL) using an ORIUS (1000) CCD 35 mm port camera.

## Image analysis
All image analysis was performed using ImageJ and FIJI (*Rueden et al., 2017*; *Schindelin et al., 2012*).

### Cytoplasm sharing calculation
Cytoplasmic sharing was quantified by manually tracing the total papillar area by morphology and the area marked by mKO2 signal in one z-slice of the papillar face of each animal. The area marked by mKO2 was summed and divided by the sum of the total papillar area to yield the papillar fraction marked by mKO2 which indicates the degree of cytoplasmic sharing within each animal. Papillae without mKO2 signal were excluded from the area measurements.

**Table 6.** Additional statistics.

| Panel | N (animals) per group | Bio. reps | Statistical test | P-value |
|---|---|---|---|---|
| *Figure 1G* | 9–18 | 2 | Unpaired t-test | 66HPPF:74HPPF < 0.0001 |
| *Figure 2H* | 9–32 | 2–3 | One-way ANOVA with Tukey's multiple comparisons test | ANOVA:<0.0001 Pre:WT < 0.0001 WT:*shi #1* < 0.0001 WT:*shi #2* < 0.0001 WT:*Rab5 #1* < 0.0001 WT:*Rab5 #2* < 0.0001 WT:*Rab11 #1* < 0.0001 WT:*Rab11 #2* < 0.0001 *shi #1*:*Rab5 #2* 0.0181 *shi #1*:*Rab11 #2* 0.0428 *shi #2*:*Rab5 #2* 0.0263 *Rab5 #1*:*Rab5 #2* 0.0009 *Rab5 #1*:*Rab11 #2* 0.0020 all others, ns |
| *Figure 3C* | 6–10 | 2–3 | see 3-S1A | see *Figure 3—figure supplement 1A* |
| *Figure 3E* | 4–5 | 3 | Unpaired t-test | Apical region: Pre:Post < 0.0001 |
| *Figure 3P* | 3–4 | 2 | Unpaired t-test | WT:*shi RNAi* < 0.0001 |
| *Figure 4B* | 3–4 | 2 | Unpaired t-test | Pre:Post < 0.0001 |
| *Figure 4F* | 13–14 | 2 | One-way ANOVA with Tukey's multiple comparisons test | ANOVA:<0.0001 WT:*ogre*$^{DN}$ < 0.0001 WT:*Df* < 0.0001 WT:*ogre RNAi* 0.0007 |
| *Figure 4H* | 27–37 | 3 | One-way ANOVA with Tukey's multiple comparisons test (mean death at 10 days in each group) | ANOVA:<0.0001 WTsalt:*shi*$^{DN}$reg ns, 0.7173 WTsalt:*shi*$^{DN}$salt < 0.0001 *shi*$^{DN}$salt:*shi*$^{DN}$reg < 0.0001 ANOVA:<0.0001 WTsalt:*ogre*$^{DN}$reg < 0.0001 WTsalt:*ogre*$^{DN}$salt < 0.0001 *ogre*$^{DN}$salt:*ogre*$^{DN}$reg < 0.0001 |
| *Figure 1—figure supplement 1H* | 12–20 | 2 | Unpaired t-test | WT:*fzr RNAi* < 0.0001 WT:*N*$^{DN}$ ns, 0.1786 |
| *Figure 2—figure supplement 1A* | 8–11 | 2 | One-way ANOVA with Tukey's multiple comparisons test | ANOVA:<0.0001 *Sing RNAi*:all others < 0.0001 All others: ns |
| *Figure 2—figure supplement 1B* | 6–8 | 2 | One-way ANOVA | ANOVA: ns, 0.3692 |
| *Figure 2—figure supplement 1C* | 11–23 | 2 | One-way ANOVA with Tukey's multiple comparisons test | ANOVA: 0.0044 *shi RNAi #1*:*Rab11 RNAi #1* 0.0244 *Rab5 RNAi #2*:*Rab11 RNAi #1* 0.0193 All others: ns |
| *Figure 2—figure supplement 1F* | 10–11 | 2 | Unpaired t-test | ns, 0.0782 |
| *Figure 3—figure supplement 1A* | 6–10 | 2 | One-way ANOVA with Tukey's multiple comparisons test | ANOVA:<0.0001 Pre:Post < 0.0001 Pre:*shi RNAi* ns, 0.7882 Post:*shi RNAi* < 0.0001 |
| *Figure 3—figure supplement 1B''* | 10 | 2 | Unpaired t-test | Apical basal difference (see 1-S3A) Pre:Post 0.0007 |
| *Figure 3—figure supplement 1G* | 3–4 | 2 | Unpaired t-test | ns, 0.2203 |
| *Figure 3—figure supplement 1G'* | 3–4 | 2 | Unpaired t-test | ns, 0.4754 |

*Table 6 continued on next page*

*Table 6 continued*

| Panel | N (animals) per group | Bio. reps | Statistical test | P-value |
|---|---|---|---|---|
| *Figure 3—figure supplement 1G''* | 3–4 | 2 | Multiple unpaired t-tests | Septate: WT:*shi RNAi* ns, 0.1547 Gap: WT:*shi RNAi* < 0.0001 |
| *Figure 4—figure supplement 1A* | 3–4 | 2 | One-way ANOVA | ns, 0.8973 |
| *Figure 4—figure supplement 1A'* | 3–4 | 2 | One-way ANOVA | ns, 0.3994 |
| *Figure 4—figure supplement 1A''* | 3–4 | 2 | Multiple unpaired t-tests | Septate: all ns Gap: Pre:Post 0.0004 Gap: all others, ns |
| *Figure 4—figure supplement 1E''* | 11 | 2 | Unpaired t-test | WT:*shi*$^{DN}$ < 0.0001 |

## Line profiles

For line profile data collection, fixed and mounted hindguts were imaged on a Zeiss Apotome on the 40Xoil objective. Once moved into ImageJ, the images were rotated with no interpolation so that the central canal was perpendicular to the bottom of the image. From the midline of the central canal, a straight line (width of 300) was drawn out to one edge of the papillae. One papilla was measured per animal. Papillae were measured at the widest width. Next, the Analyze > Plot Profile data was collected from this representative 300 width line and moved into Excel. In Excel, the data was first was normalized to the maximum length of the papillae and the maximum GFP intensity per animal. Each data point is a % of the total length of the papillae and a % of the maximum GFP intensity. Next, the X values were rounded to its nearest 1% value. Next, all the Y-values were averaged per X value bins (average % GFP intensity per rounded % distance value). % GFP intensity values were plotted from 1–100% total distance of papilla.

## Statistical analysis

Statistical analysis was performed in GraphPad Prism 8. Detailed statistical tests and methods are reported in *Table 6*.

## Genotype and experiment-specific method notes

Some additional methodological details, including animal genotype, applied to only a specific figure panel. Please see *Table 6* for this information.

## Acknowledgements

We thank members of the Fox laboratory and Drs. Dong Yan and Tony Harris for valuable feedback. Ying Hao (Duke Eye Center) provided assistance with electron microscopy. The Duke Light Microscopy Core Facility supplied training and microscopes that were used for live and fixed fluoresecence microscopy. Jamie Roebuck (Duke University) generated the transgenic *UAS-Gapdh2-GFP*$^{PA}$ flies.

## Additional information

### Funding

| Funder | Grant reference number | Author |
|---|---|---|
| National Institutes of Health | GM118447 | Donald T Fox |
| National Institutes of Health | HL140811 | Nora G Peterson |

The funders had no role in study design, data collection and interpretation, or the decision to submit the work for publication.

### Author contributions

Nora G Peterson, Juliet S King, Conceptualization, Resources, Data curation, Software, Formal analysis, Supervision, Funding acquisition, Validation, Investigation, Visualization, Methodology, Writing - original draft, Project administration, Writing - review and editing; Benjamin M Stormo, Conceptualization, Resources, Data curation, Formal analysis, Validation, Investigation, Visualization, Methodology; Kevin P Schoenfelder, Conceptualization, Resources, Data curation, Software, Formal analysis, Validation, Investigation, Visualization, Methodology; Rayson RS Lee, Resources, Data curation, Formal analysis, Validation, Investigation, Visualization; Donald T Fox, Conceptualization, Supervision, Funding acquisition, Visualization, Writing - original draft, Project administration, Writing - review and editing

### Author ORCIDs

Nora G Peterson (iD) https://orcid.org/0000-0002-7734-1861
Benjamin M Stormo (iD) http://orcid.org/0000-0002-6861-8451
Donald T Fox (iD) https://orcid.org/0000-0002-0436-179X

### Decision letter and Author response

Decision letter https://doi.org/10.7554/eLife.58107.sa1
Author response https://doi.org/10.7554/eLife.58107.sa2

## Additional files

### Supplementary files

- Transparent reporting form

### Data availability

All data generated or analyzed during this study are included in the manuscript and supporting files.

The following previously published dataset was used:

| Author(s) | Year | Dataset title | Dataset URL | Database and Identifier |
|---|---|---|---|---|
| Leader DP, Krause SA, Pandit A, Davies SA, Dow JAT | 2018 | FlyAtlas2 | http://flyatlas.gla.ac.uk/FlyAtlas2/index.html | FlyAtlas, 10.1093/nar/gkx976 |

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
