## [Decision Letter]

**Acceptance summary:**

This report of a mechanism for the sharing of cytoplasmic contents in multinucleated cells was both distinct from previously reported mechanisms of failed cytokinesis and plasma membrane breaches and of interest to a broad readership. The authors use an elegant approach for identifying cells with shared cytoplasms by screening brainbow flies for the appearance of cells with "mixed" labels. The current mechanisms of cytoplasmic sharing are scant, and this paper goes quite a distance in rectifying this without the additional gap junction studies.

**Decision letter after peer review:**

Thank you for submitting your article "Cytoplasmic sharing through apical membrane remodeling" for consideration by *eLife*. Your article has been reviewed by three peer reviewers, one of whom is a member of our Board of Reviewing Editors, and the evaluation has been overseen by Utpal Banerjee as the Senior Editor. The following individual involved in review of your submission has agreed to reveal their identity: Yukiko M Yamashita (Reviewer #3).

The reviewers have discussed the reviews with one another and the Reviewing Editor has drafted this decision to help you prepare a revised submission.

Summary:

The manuscript by Petersen and colleagues reports a novel mechanism for the sharing of cytoplasmic contents in multinucleated cells. This mechanism is seemingly distinct from previously reported mechanisms of failed cytokinesis and plasma membrane breaches. The authors use an elegant approach for identifying cells with shared cytoplasms by screening brainbow flies for the appearance of cells with "mixed" labels. They both identify a novel population in the *Drosophila* rectal papillae, and go on to use this approach to screen for genes essential for the mixing phenotype. Once they fail to find hits for canonical actin-based cytoskeletal remodelling, they go on to conduct secondary screens for endocytosis and membrane trafficking factors, which are critical for cytoplasmic sharing. Finally, they examine cellular ultrastructure by EM and fail to find evidence of plasma membrane breaches, consistent with their genetic manipulations, but they do observe pronounced apical membrane remodelling and the presence of gap junctions coinciding with the time of sharing. Pursing this avenue further, they genetically perturb gap junction proteins, and while not as significant as membrane remodelling factors (likely due to their redundancy), the authors demonstrate that gap junctions are important for cytoplasmic sharing. This process of cytoplasmic sharing seems physiologically important, as flies with failed sharing are not able to cope with a high-salt diet challenge and die.

Reviewer #1:

This work is interesting and broadly relevant, as mechanisms of cytoplasmic sharing are scant, although the presence of multinucleated cells is common throughout the animal kingdom. The work is elegant, well controlled, and accurately described. In particular, the authors should be praised for their clear communication of the results and their implications. Although some comments need to be addressed, which are listed below, the paper is a good fit for *eLife* and warrants publication pending some minor changes to the manuscript.

1) The paragraph beginning "We next examined whether cytoplasm sharing requires the distinctive papillar cell cycle program, which completes prior to sharing onset (Figure 1—figure supplement 1D). Larval papillar…" is hard to understand for a general reader. Can the authors better place their current results in the context of their previous studies (Fox et al, 2010)?

2) The authors conduct photo-activation experiments to show that protein at least the size of GFP (~27kDa) can be shared between multi-nucleated regions. This extremely interesting observation suggests that large macromolecules may pass through gap junctions. Further data to test this would be interesting, including conducting photo-activation/macromolecule mobility experiments with macromolecules of different size and physicochemical properties (e.g. fluorescent proteins with different size or charge, e.g. tdTomato, supercharged-GFP).

3) The quality of EM images should be improved. This may be mostly due to reproduction in the.pdf, however some of the false coloring could be adjusted and labelled better to understand what the authors are trying to communicate. Cartoons of the orientation and region of cell/tissue being imaged would also help.

4) The following sentence appears to be an oxymoron "a straight to a more tortuous morphology around the time of cytoplasm sharing onset".

5) Further speculation of how ion transport in the gut may be affected by a lack of cytoplasmic sharing would be interesting, in addition to their discussion of the potential role of formation of intracellular membrane stacks. Is there evidence for transluminal transport affected by cytoplasmic properties in other systems? Why might this be advantageous for the animal?

6) In general, a clearer cartoon/schematic of the rectal papilla as well as the experimental flow would be helpful in the main figure, especially for readers without expertise in *Drosophila* models. It is a bit difficult to discern exactly when the heat-shock to induce Cre expression was being carried out, especially in the case that the Gal80ts fly line was also being used to repress Gal4 expression. Perhaps this information could be added to the timeline in Figure 1—figure supplement 1D and included in a main figure rather than in the supplement.

7) The data on localization of Rab5 endosomes could be strengthened. It would be nice to see other markers, that don't rely on over-expression of a transgenic construct. Alternatively, could the authors also assess from their EM data whether they see redistribution of vesicles pre- and post- sharing?

Reviewer #2:

In this manuscript, Peterson et al. describe cytoplasmic share across a large number of cells in *Drosophila* rectal papillae. Employing the dBrainbow system, the authors identify the rectal papilla as a novel tissue that undergoes cytoplasmic sharing. They show that this is a regulated process occurring 68 hours post puparium formation. Interestingly, none of the proteins involved in myoblast fusion seem to be essential for cytoplasmic sharing in the rectal papillae. Instead, the authors show that various proteins involved in vesicle trafficking are necessary for cytoplasm sharing. In particular, they implicate a role for the membrane vesicle recycling circuit consisting of Shibire, Rab5 and Rab11 in the process of cytoplasmic sharing. Knockdown of these components leads to defective cytoplasmic sharing. Furthermore, the authors show that cytoplasmic sharing is accompanied by extensive membrane reorganization. Electron micrographs reveal that cytoplasmic sharing is not accompanied by any membrane breaches but rather formation of gap-junction like structures. Knockdown of the gap-junction proteins, the Inxs, results in defects in cytoplasmic sharing, further supporting a role for gap junctions in the process. It is interesting to note that animals defective in cytoplasmic sharing are intolerant of a high-salt diet implicating a physiological role for this process during development.

This is an interesting study that identifies a novel tissue undergoing cytoplasmic sharing in the absence of plasma membrane breaching. The identification of Shi, Rab5, Rab11 and gap junction proteins in this process based on their mutant phenotypes is also intriguing, although the mechanisms by which these proteins promote cytoplasmic sharing remain unclear.

Specific Comments:

1) The authors made interesting observations that massive membrane reorganization in apical microvilli-like structures, apical cell–cell junctions, and endomembrane stacks surrounding mitochondria coinciding with cytoplasm sharing. However, it is unclear how/which of these membrane reorganization events would lead to cytoplasmic sharing.

2) The authors showed the appearance of gap junctions during cytoplasmic sharing. Do the plasma membranes for these gap junctions come from exocytosis? If so, why would inhibiting an endocytosis protein Shi inhibit gap junction?

3) Increased gap junctions and endomembrane stacks are quite separate observations. Is there any connection between these membrane structures, e.g. similar origin?

4) Could proteins larger than GFP pass through the papillar cells, presumably through the gap junctions?

5) What is the physiological significance of the association between endomembrane stacks and mitochondria?

6) How does the change of Shi localization (Figure 3D and D') contribute to cytoplasmic sharing?

7) Are Inx1-3 expressed in the Shi, Rab5 and Rab11 knockdown animals which show defective cytoplasmic sharing? If yes, is their localization altered?

8) Does *fzr* knockdown affect the levels of Shi, Rab5 and Rab11?

9) Do *fzr* mutants phenocopy the membrane architecture observed in the shi mutants?

10) What does the endosomal distribution look like in the rab5 and rab11 mutants at the onset of cytoplasmic sharing? (68HPPF)

Reviewer #3:

This study by Peterson et al. reveals a novel mechanism of cytoplasmic sharing through apical membrane remodeling (through gap junction, which allows for sharing of large molecules, much larger (>27kD) than canonical gap junction dependent diffusion (<1kD)) in *Drosophila* rectal papillae. They find that membrane trafficking pathway involving Shi, Rab5, Rab11 are involved in cytoplasmic sharing. This is a novel discovery on interesting biology of cytoplasmic sharing, with likely physiological relevance (as inhibiting cytoplasmic sharing leads to high-salt diet sensitivity). Overall, this is a high quality study that provides a novel biology: but I do have several concerns to be addressed.

Recombination can happen independently to sister chromatids if cells are in G2 phase when the recombination was induced. Plus, rectal papillae cells are polyploid, which increases the chance that a single cell can have multiple recombination events). If so, there should be a considerable number of cells that express multiple colors without cytoplasmic sharing. How do they access this possibility? Was recombination titrated such that recombination can happen only to one chromatid? The image in Figure 1D is so clear, so I don't doubt that each cell is labelled with one color, but if you think about the logic, we have to wonder why. Some discussion/explanation is necessary.

Figure 3: formation of endomembrane surrounding mitochondria during cytoplasmic sharing is interesting, but is there any evidence that this indeed contributes to cytoplasmic sharing? It's unclear how endomembrane would lead to cytoplasmic sharing.

---

## [Author Response]

Reviewer #1:1) The paragraph beginning "We next examined whether cytoplasm sharing requires the distinctive papillar cell cycle program, which completes prior to sharing onset (Figure S1D). Larval papillar…" is hard to understand for a general reader. Can the authors better place their current results in the context of their previous studies (Fox et al, 2010)?

Thank you for this opportunity to clarify. In response, we added clarifying text to provide more context to this paragraph and have moved the timeline from the supplemental figure to the main figure:

The text previously read:

“We next examined whether cytoplasm sharing requires the distinctive papillar cell cycle program, which completes prior to sharing onset (Figure 1—figure supplement 1D). Larval papillar cells first undergo endocycles, which increase cellular ploidy, and then pupal papillar cells undergo polyploid mitotic cycles, which increase cell number (Fox et al., 2010).”

The text now reads:

“We next examined whether cytoplasm sharing requires either programmed endocycles or mitoses. We have previously shown that larval papillar cells first undergo endocycles, which increase cellular ploidy, and then pupal papillar cells undergo polyploid mitotic cycles, which increase cell number (Fox et al., 2010). Both endocycles and mitoses occur well prior to the start of papillar cytoplasm sharing (Figure 1E). Papillar endocycles require the Anaphase-Promoting Complex/Cyclosome regulator *fizzy-related* (*fzr*) while the papillar mitoses require Notch signaling (Schoenfelder et al., 2014).”

2) The authors conduct photo-activation experiments to show that protein at least the size of GFP (~27kDa) can be shared between multi-nucleated regions. This extremely interesting observation suggests that large macromolecules may pass through gap junctions. Further data to test this would be interesting, including conducting photo-activation/macromolecule mobility experiments with macromolecules of different size and physicochemical properties (e.g. fluorescent proteins with different size or charge, e.g. tdTomato, supercharged-GFP).

Thank you for your interest- in response, we designed and made a transgenic UAS-Gapdh2-GFP^photoactivatable (PA)^ fly line in order to test the sharing of a larger protein. We chose Gapdh2 as it is endogenously expressed in the rectal papillae, is relatively large (35.4 kDa), and cytosolic. We used this transgenic fly to test whether a protein more than twice the size of GFP^PA^ alone can be shared between papillar cells. We found that the 62.3 kDa Gapdh2-GFP^PA^ protein is shared between papillar cells, though much more slowly as would be expected for a larger protein. We never observe it to stop at a cell–cell boundary.

3) The quality of EM images should be improved. This may be mostly due to reproduction in the.pdf, however some of the false coloring could be adjusted and labelled better to understand what the authors are trying to communicate. Cartoons of the orientation and region of cell/tissue being imaged would also help.

We apologize for the image quality issue. We believe that the file conversion and compression decreased the image quality of the EM images. We have now included higher resolution images and that should improve the image quality in the final pdf.

4) The following sentence appears to be an oxymoron "a straight to a more tortuous morphology around the time of cytoplasm sharing onset".

We have re-written this sentence to address this comment.

The text now reads:

“Just basal to the microvilli, apical cell–cell junctions are straight in early pupal development and compress into a more curving, tortuous morphology around the time of cytoplasm sharing onset.”

5) Further speculation of how ion transport in the gut may be affected by a lack of cytoplasmic sharing would be interesting, in addition to their discussion of the potential role of formation of intracellular membrane stacks. Is there evidence for transluminal transport affected by cytoplasmic properties in other systems? Why might this be advantageous for the animal?

Thank you for the opportunity to further speculate on this topic. At this point, we truly can only speculate.

In the previous manuscript version, we hypothesized:

“We speculate that papillar cytoplasm movement across a giant multinuclear structure enhances resorption by facilitating interaction of ions and ion transport machinery with intracellular membrane stacks.”

In the revised version, we expand upon this hypothesis. The revised text now states:

“Arthropod papillar structures are subject to peristaltic muscle contractions from an extensive musculature (Rocco et al., 2017), which aid in both excretion and movement of papillar contents into the hemolymph (Mantel, 1968). Further, relative to other hindgut regions, the rectum appears to have specialized innervation and regulation by the kinin family of neuropeptides, which are hypothesized to provide additional input in to muscle activity in this critical site of reabsorption (Audsley and Weaver, 2009, Lajevardi and Paluzzi, 2020). We speculate that these muscle contractions aid in vigorous movement of papillar cytoplasm, which includes ions and water taken up from the intestinal lumen. The movement of these papillar contents may facilitate both cytoplasm exchange between papillar cells and the interaction of ions and ion transport machinery with intracellular membrane stacks.”

Regarding the reviewer’s question about transluminal transport and the advantage to the animal, these remain open questions that we look forward to addressing in the future.

6) In general, a clearer cartoon/schematic of the rectal papilla as well as the experimental flow would be helpful in the main figure, especially for readers without expertise in Drosophila models. It is a bit difficult to discern exactly when the heat-shock to induce Cre expression was being carried out, especially in the case that the Gal80ts fly line was also being used to repress Gal4 expression. Perhaps this information could be added to the timeline in Figure S1D and included in a main figure rather than in the supplement.

We thank the reviewer for this suggestion. We moved the timeline in Figure 1—figure supplement 1 to Figure 1, modified the timeline to include Cre, and clarified Cre and Gal4-expression in writing.

The text now reads:

“We used animals heterozygous for *UAS-dBrainbow* to ensure single-labeling of cells. We ubiquitously expressed *Cre,* which does not require heat-shock induction, from early embryonic stages, before cells endocycle to any great degree. Cre-mediated excision occurs independently of Gal4 expression and Gal80^ts^ repression of dBrainbow. Therefore, we can ensure that multi-labeled cells only arise by cytoplasm sharing between cells not related by cell division or incomplete cytokinesis.”

We have also added a small diagram of a papilla to Figure 1 (Figure 1D).

7) The data on localization of Rab5 endosomes could be strengthened. It would be nice to see other markers, that don't rely on over-expression of a transgenic construct. Alternatively, could the authors also assess from their EM data whether they see redistribution of vesicles pre- and post- sharing?

Thank you for this opportunity to discuss and address our endosome localization data further. In the revised manuscript, we address this point by clarifying and emphasizing the GFP-myc-FYVE marker is not, in fact, an overexpression of a protein that would affect endosome localization with the following additional text:

**“**GFP-tagged pan-endosome marker (*myc-2x-FYVE*), overexpression of which should not alter endosome shape or localization (Gillooly et al., 2000, Wucherpfennig et al., 2003),”

Further, in response to this comment, we also used a Rab5 antibody to show Rab5 localization without using transgenic markers. The Rab5 antibody does not mark full endosomes and has a punctate appearance in papillar cells, as shown In Author response image 1. As it does not mark full endosomes, we do not observe the same degree of polarization observed with the GFP-myc-FYVE and Rab5-GFP transgenes. This is in agreement with other literature (Wucherpfennig et al., 2003). We also note that we cannot look at the same time point with the antibody as with transgenes, as a thick cuticle layer forms on papillae in late development, which causes technical issues with antibody staining. We therefore use an earlier timepoint soon after cytoplasm sharing instead.

Reviewer #2:Specific Comments:1) The authors made interesting observations that massive membrane reorganization in apical microvilli-like structures, apical cell–cell junctions, and endomembrane stacks surrounding mitochondria coinciding with cytoplasm sharing. However, it is unclear how/which of these membrane reorganization events would lead to cytoplasmic sharing.

Thank you for the opportunity to expand on this point, which falls beyond the scope of the current manuscript. The reviewer is absolutely correct- at this time it is unclear to us how the multiple membrane reorganization events which we report here to be directed by Dynamin, Rab5, and Rab11, are inter-related. We show here that they all occur within a succinct developmental window, in conjunction with the relocalization of Dynamin and endosomes. We speculate that each reorganization event is critical to transform this epithelium into a highly specialized reabsorptive structure. Future work can hopefully identify separation of function mutants that perturb only specific aspects of papillar membrane remodeling, so that we can individually evaluate the contribution of each to cytoplasmic sharing and papillar physiology. We note that a similar reviewer comment from a 2014 publication led us to first consider cytoplasm sharing as a possibility in papillar cells. We really value such input, and we absolutely intend to pursue these questions in the future!

2) The authors showed the appearance of gap junctions during cytoplasmic sharing. Do the plasma membranes for these gap junctions come from exocytosis? If so, why would inhibiting an endocytosis protein Shi inhibit gap junction?

This is an interesting question that we hope to answer in future studies. We speculate that plasma membrane and septate junction that exists in the apical region prior to sharing is sculpted and perhaps removed by endocytic factors. If this region is not remodeled, then there is no place for gap junction establishment. This is supported by staining of Inx3, a gap junction protein, that appears to be expressed in *shi* RNAi animals (Figure 4—figure supplement 1C-C’) but does not localize to cell–cell boundaries. However, we cannot entirely rule out that Shi has an indirect effect on gap junction establishment, such as through papillar cell differentiation or signaling.

3) Increased gap junctions and endomembrane stacks are quite separate observations. Is there any connection between these membrane structures, e.g. similar origin?

We agree with the reviewer. This comment is highly related to the above comment #1 from reviewer 2. Please see our response to that comment regarding this question, which is beyond the scope of our current manuscript.

4) Could proteins larger than GFP pass through the papillar cells, presumably through the gap junctions?

Please see our response to comment #2 from reviewer #1.

5) What is the physiological significance of the association between endomembrane stacks and mitochondria?

Thank you for the opportunity to address this interesting question. In response, we added text to expand on the significance of mitochondrion-endomembrane stack association. Berridge and Gupta, 1967, hypothesized that the mitochondria provide ATP to active ion transport ATPases. Patrick et al. (2006) found P-type Na+/K^+^-ATPase localized to the basal edge of *Aedes aegypti* rectal pads. We hypothesize that mitochondria supply ATP to P-type Na+/K^+^-ATPase among other ATP-dependent ion transporters located in the endomembrane stacks. The endomembrane stacks and associated mitochondria therefore support ion recycling from the rectal lumen back into the hemolymph.

6) How does the change of Shi localization (Figure 3D and D') contribute to cytoplasmic sharing?

Thank you for this question. At this time, we do not know if the change in Shi localization directly contributes to cytoplasmic sharing. We do show that Shi is required for changes in endosome positioning but that knock down of Rab5 does not affect Shi localization which suggests that Shi localization is upstream of the endosome positioning that occurs concurrently with cytoplasm sharing. This is certainly something to explore in the future.

7) Are Inx1-3 expressed in the Shi, Rab5 and Rab11 knockdown animals which show defective cytoplasmic sharing? If yes, is their localization altered?

Thank you for your interest- in response, we stained *shi* knockdown animals with anti-Inx3 antibody and found that Inx3 does not localize to cell–cell boundaries as in age-matched WT animals, which is consistent with our *shi* RNAi electron micrographs. We added Panel C-C’ to Figure 4—figure supplement 1 and the following text:

“Inx3 also does not localize to cell–cell boundaries in *shi* RNAi animals (Figure 4—figure supplement 1C-C’).”

We note that we looked at localization but not overall protein levels (by Western blot, for example) due to very limited antibody, so at this time we cannot conclude if *shi* RNAi affects Innexin expression as well as Innexin localization.

8) Does fzr knockdown affect the levels of Shi, Rab5 and Rab11?

Thank you for your interest- in response, we stained post-sharing WT and *fzr* RNAi animals for Rab5 and found that Rab5 looks similar in localization and level in WT and *fzr* RNAi animals, as shown In Author response image 2. This suggests that *fzr* RNAi is not acting directly through Rab5.

**Author response image 2. respfig2:** 

9) Do fzr mutants phenocopy the membrane architecture observed in the shi mutants?

We share the reviewer’s interest in this question. However, due to COVID-19, we do not have regular access to the EM facility, and therefore we cannot address this point in a timely manner. We do note that we have previously shown that *fzr* RNAi blocks papillar endocycles in larval development, and therefore we speculate that these endocycles are important for papillar cell identity and differentiation. As such, we would expect that the membrane architecture in *fzr* RNAi animals is not wild-type. However, given how early the endocycles occur in development, we expect that adult *fzr* RNAi animals have ultrastructure similar to larval papillar cells while *shi* RNAi animals are more like WT adult cells with grossly impaired membrane reorganization.

10) What does the endosomal distribution look like in the rab5 and rab11 mutants at the onset of cytoplasmic sharing? (68HPPF)

This is certainly an interesting question. In response, we used a Rab5 antibody to examine early endosomes and did not see an obvious difference in Rab5 distribution between WT and Rab11 RNAi animals around the time of sharing (not shown). The caveat is that we used Rab5 staining instead of the GFP-Myc-2x-FYVE or Rab5-YFP overexpression to mark endosomes. This is certainly a question to explore in more detail in the future.

Reviewer #3:[…] Recombination can happen independently to sister chromatids if cells are in G2 phase when the recombination was induced. Plus, rectal papillae cells are polyploid, which increases the chance that a single cell can have multiple recombination events). If so, there should be a considerable number of cells that express multiple colors without cytoplasmic sharing. How do they access this possibility? Was recombination titrated such that recombination can happen only to one chromatid? The image in Figure 1D is so clear, so I don't doubt that each cell is labelled with one color, but if you think about the logic, we have to wonder why. Some discussion/explanation is necessary.

Please see our response to reviewer #1, comment #6.

Figure 3: formation of endomembrane surrounding mitochondria during cytoplasmic sharing is interesting, but is there any evidence that this indeed contributes to cytoplasmic sharing? It's unclear how endomembrane would lead to cytoplasmic sharing.

Please see our response to comment #1 from reviewer #2.